# Combining the Peptide RWQWRWQWR and an Ethanolic Extract of *Bidens pilosa* Enhances the Activity against Sensitive and Resistant *Candida albicans* and *C. auris* Strains

**DOI:** 10.3390/jof9080817

**Published:** 2023-08-02

**Authors:** Yerly Vargas-Casanova, Claudia Patricia Bravo-Chaucanés, Andrea Ximena Hernández Martínez, Geison Modesti Costa, Jorge Luis Contreras-Herrera, Ricardo Fierro Medina, Zuly Jenny Rivera-Monroy, Javier Eduardo García-Castañeda, Claudia Marcela Parra-Giraldo

**Affiliations:** 1Microbiology Department, Faculty of Sciences, Pontificia Universidad Javeriana, Bogotá 110231, Colombia; y.vargasc@javeriana.edu.co (Y.V.-C.); claub06@gmail.com (C.P.B.-C.); 2Chemistry Department, Faculty of Sciences, Pontificia Universidad Javeriana, Bogotá 110231, Colombia; hernandez_a@javeriana.edu.co (A.X.H.M.); modesticosta.g@javeriana.edu.co (G.M.C.); 3Faculty of Engineering and Basic Sciences, Insitituto Tecnológico del Putumayo, Mocoa 860001, Colombia; jorge.contreras@itp.edu.co; 4Faculty of Sciences, Universidad Nacional of Colombia, Bogotá 111321, Colombia; rfierrom@unal.edu.co (R.F.M.); zjriveram@unal.edu.co (Z.J.R.-M.); jaegarciaca@unal.edu.co (J.E.G.-C.)

**Keywords:** bovine lactoferricin peptides, *Bidens pilosa*, synergistic activity, *Candida albicans*, *C. auris*

## Abstract

The antifungal activity of palindromic peptide RWQWRWQWR and its derivatives was evaluated against clinical isolates of *Candida albicans* and *C. auris*. Also, *Bidens pilosa* ethanolic extracts of leaves and stem were evaluated. Furthermore, combinations of peptide, extract, and/or fluconazole (FLC) were evaluated. The cytotoxicity of peptides and extracts in erythrocytes and fibroblasts was determined. The original palindromic peptide, some derivative peptides, and the ethanolic extract of leaves of *B. pilosa* exhibited the highest activity in some of the strains evaluated. Synergy was obtained between the peptide and the FLC against *C. auris* 435. The combination of the extract and the original palindromic peptide against *C. albicans* SC5314, *C. auris* 435, and *C. auris* 537 decreased the minimal inhibitory concentrations (MICs) by a factor of between 4 and 16. These mixtures induced changes in cell morphology, such as deformations on the cell surface. The results suggest that the combination of RWQWRWQWR and *B. pilosa* extract is an alternative for enhancing antifungal activity and decreasing cytotoxicity and costs and should be considered to be a promising strategy for treating diseases caused by *Candida* spp.

## 1. Introduction

Natural products (NPs) have been broadly defined as chemicals produced by living organisms such as plants, animals, and microorganisms. NPs can be classified into primary and secondary metabolites, where the latter are considered important for the organism since they provide an evolutionary advantage, acting as “chemical protection agents” against predators, pathogens, or competing organisms [1,2]. For that reason, historically NPs have been used in the therapeutic field, especially in cancer and infectious diseases [3].

Antimicrobial peptides (AMPs) [4,5,6] and metabolites from plant extracts are examples of NPs generated for the defense of host animals, plants, and microorganisms [3,7,8,9]. In mammals, AMPs can be found in epithelial cells, granulocytes, and body fluids such as milk and tears [10,11,12]. Bovine lactoferricin (LfcinB) is an AMP derived from bovine lactoferrin (LFB) [13] that exhibits antimicrobial activity, including against fungi such as *Candida albicans* and dermatophytes [14,15,16]. Also, it has been seen that its short derived peptides have equal or greater activity than LfcinB and LFB against different species of *Candida* [17,18,19] and can have a synergistic effect when evaluated in combination with conventional antifungals [20].

In a previous study [21], we evaluated the antifungal activity of a palindromic peptide derived from LfcinB, LfcinB (21–25)_Pal_: RWQWRWQWR, against strains and clinical isolates of *C. albicans*, *C. glabrata*, *C. krusei*, *C. auris*, and *C. tropicalis*. This peptide exhibited activity against all the yeasts evaluated, with minimal inhibitory concentrations (MICs) between 50 and 200 μg/mL. Additionally, it was fungistatic and fungicidal after 48 h of incubation. Finally, LfcinB (21–25)_Pal_ (25–50 μg/mL) combined with fluconazole (FLC) exhibited a synergistic antifungal effect against *C. tropicalis* 883 and *C. krusei* 6258 (resistant to FLC). The results of this study suggested that the palindromic peptide could have a therapeutic application alone or in combination with FLC, it being of interest to explore other types of combinations.

With respect to plants, chemical defense compounds have been found, some of which exhibit antimicrobial activity against microorganisms of importance in the clinical field [22]. In this context, *Bidens pilosa* is a plant species recognized in traditional medicine; *B. pilosa* belongs to the *Asteraceae* family, has worldwide distribution, and grows in temperate and tropical zones. Multiple biological functions have been reported that do not induce significant adverse effects, showing evidence of their phytochemical and biological relevance [23]. Extracts of *B. pilosa* (leaves, stems, flowers, and seeds) with organic solvents (hexane, dichloromethane, ethyl acetate, and methanol) exhibited antifungal activity, and methanol extracts of the leaves acted against reference strains and clinical isolates of *Candida* spp. [24,25,26,27,28].

Epidemiological studies indicate that there has been an increase in the incidence of fungal infections, such as invasive fungal disease (IFD), where *C. albicans* and *C. auris* are etiological agents [29,30]. This increase is mainly due to the use of invasive therapies, the presence of diseases that compromise the immune system, such as acquired immunodeficiency syndrome (AIDS) and cancer, the indiscriminate use of antifungal agents and prophylactic antibiotics, and the emergence of the coronavirus disease 2019 (COVID-19) pandemic [29,30,31,32,33,34]. For the treatment of diseases associated with *Candida* spp., a limited number of antifungal agents are used that have a degree of toxicity and resistance [35,36,37,38]. There is a need to find new antifungal agents, so AMPs and *B. pilosa* extracts may be an interesting alternative [3,39,40,41,42,43,44].

The palindromic peptide RWQWRWQWR has exhibited activity against Gram-positive and Gram-negative strains, *C. auris* and *C. albicans* strains, breast cancer cells, colon cancer cells, cervix cancer cells, etc. This molecule is considered to be promising for developing treatments against bacterial and fungal infections, as well as cancer [21,45,46,47,48,49]. Peptides derived from palindromic peptides containing non-natural amino acids exhibited similar or greater anticancer activity than the original peptide [50].

Pharmacological activity has been described for *B. pilosa*, which includes the following effects: anticancer, antidiabetic, anti-inflammatory, immunomodulatory, antioxidant, antimalarial, antibacterial, wound treatment, hypotensive, and antifungal [23,27]. Ethanolic and methanolic extracts of leaves generated halos of between 6 and 8 mm (250–1250 µg/mL) and 40 mm (1.562–50 µg/mL), respectively, against *C. albicans* [24,26]. In 2021, Angelini et al. reported antifungal activity against sensitive and resistant *Candida* spp. using a methanolic extract of the leaves. The extract was composed mainly of phenolic acids, to which antifungal activity has been attributed [28].

Due to the above, the antifungal activity of RWQWRWQWR, peptides derived from this sequence, and extracts of *B. pilosa* were evaluated in reference strains and sensitive and resistant clinical isolates of *C. albicans* and *C. auris*.

## 2. Materials and Methods

Roswell Park Memorial Institute (RPMI) 1640 medium (Sigma-Aldrich, St. Louis, MA, USA) (with MOPS, pH 7.2), dimethylsulfoxide (DMSO), Sabouraud dextrose agar (SDA; Difco), saline solution (SS), iMarKTM microplate reader (Bio-Rad, Hercules, CA, USA), FLC, Bioscreen C equipment, Bioscreen software (Growth Curves USA, Piscataway, NJ, USA), 100-well microplates (honeycomb), LIVE/DEAD™ Yeast Viability kit (Thermo Fisher Scientific, Waltham, MA, USA), Olympus FV 1000 confocal microscope, Tescan Lyra 3 microscope, Fmoc-Arg(Pbf)–OH, Fmoc-Trp(Boc)–OH, Fmoc-Gln(Trt)–OH, Fmoc-Leu-OH, Fmoc-Lys(Fmoc)–OH, Fmoc-6-Ahx-OH, Rink amide resin, dicyclohexilcarbodiimide (DCC), and 1-hydroxy-6-chlorobenzotriazole were purchased from AAPPTec (Louisville, KY, USA). Trifluoroacetic acid (TFA), acetonitrile (ACN), dichloromethane (DCM), *N*,*N*-dimethylformamide, ethanodithiol, triisopropylsilane, methanol, acetonitrile, and isopropanol were obtained from Merck (Darmstadt, Germany). SPE SupelcleanTM columns were purchased from Sigma-Aldrich (St. Louis, MO, USA).

### 2.1. Peptides

Peptides were synthesized via solid-phase peptide synthesis using the Fmoc/tBu strategy, purified using RP-SPE chromatography, and characterized by means of RP-HPLC and MS, following the protocol reported by [21,48,49,50].

### 2.2. Plant Material and Extract Preparation

Individuals of the species *B. pilosa* were collected in November 2021, in the municipality of Mocoa, Putumayo, Colombia; geographic coordinates: latitude: 1.17512521 N, longitude −76.65982164. The taxonomic determination of the species was carried out in the herbarium of the Instituto Tecnológico del Putumayo. The plant material was washed with 5% hypochlorite and then dried in an oven with circulating air at 40 °C for 72 h (soft plant material, e.g., leaves) or 96 h (hard plant material, e.g., stem). Next, the dry material was subsequently ground in a blade mill and subjected to the following extractive processes: (i) extraction by percolation with EtOH 96% at a ratio of 1:10 (*w*/*v*), at room temperature, protected from light in 4 cycles of 24 h each (solvent changes); (ii) extraction by maceration, carried out in a way similar to the previous procedure, but the cycles extended to 72 h each. The extracts from the different cycles were pooled and concentrated under reduced pressure by rotary evaporation at a temperature not exceeding 40 °C and stored at room temperature in duly labeled amber vials for later analysis [51].

A stock solution of *B. pilosa* ethanolic extract was dissolved in dimethyl sulfoxide (DMSO) at a final concentration of 80 mg/mL and stored at −20 °C. In all experiments, a DMSO control was included.

### 2.3. Phytochemical Analysis

#### Chromatography Analysis

A chemical profile was obtained using ultra-high-performance liquid chromatography coupled with photodiode array detection (UPLC-PAD) on an ACQUITY UPLC system (Waters, Milford, MA, USA), which consisted of a photodiode array detector, quaternary pump, degasser, column oven, and autosampler. The separation of metabolites was performed using Phenomenex^®^ Kinetex C18 100 Å (75 mm × 2.1 mm; 2.6 µm). The mobile phase was formic acid 0.1% (component A) and acetonitrile (component B) as follows: 0–8 min, 16% B; 8–10 min, 16–20% B; 10–12 min, 20% B; 12–15 min, 20–22% B; 15–18 min, 22–27% B; 18–20 min, 27–30% B; 20–23 min, 30–35% B; 23–28 min, 35% B; 28–34 min, 35–90% B; and then back to 16% B in 1 min [52]. The samples were injected automatically at 3 µL. The column temperature was maintained at 30 °C. The flow rate was set at 0.4 mL/min. The photo diode array (PDA) detection was set at 340 nm. Ultra-high-performance liquid chromatography–electrospray ionization–quadrupole time-of-flight–mass spectrometry (UHPLC-ESI-QToF-MS) analysis was performed on Nexera LCMS 9030 Shimadzu Scientific-Instruments (Columbia, MD, USA) equipment. The chromatographic conditions were the same as those described previously: mass spectrum detection for negative ion mode. The capillary potentials were set at +3 kV, drying gas temperature 250 °C, and the flow rate of drying gas 350 L/min.

### 2.4. Fungal Strains

In this study, the reference strain *C. albicans* SC5314, along with a *C. albicans* 256-PUJ-HUSI, FLC-resistant isolate, and two clinical isolates of *C. auris* 435-PUJ-HUSI, FLC-sensitive, and *C. auris* 537-PUJ-HUSI, FLC and amphotericin B (AmB)-resistant, previously characterized and identified (matrix-assisted laser desorption/ionization–time-of-flight (MALDI TOF)-MS) in our group, was used [21,53]. Subsequently, strains were grown on SDA plates and incubated overnight at 37 °C and stored at 4 °C for further use.

### 2.5. Antifungal Activity Assays

For the determination of the MICs, the guidelines of the Clinical Laboratory Standards Institute (CLSI BMD-M27-A4) were used with some modifications [54,55,56]. Serial dilutions were made, for peptides from 6.25 to 200 µg/mL and for extracts from 31.2 to 2000 µg/mL, with RPMI 1640 medium in a 96-well microdilution plate (100 µL), and then 100 µL of the adjusted inoculum (0.5–2.5 × 10^3^ cells/mL) was added and incubated at 37 °C, and visual and spectrophotometric reading (595 nm) was performed at 48 h. Controls: growth control in RPMI or DMSO, control with a known antifungal: FLC, *C. albicans* SC5314 (1 µg/mL), *C. albicans* 256 (64 µg/mL), *C. auris* 435 (8 µg/mL), and *C. auris* 537 (128 µg/mL), sterility controls correspond to RPMI medium and SS. The MIC was defined as the minimum concentration of peptide or extract that inhibited 50% of growth compared to the control. To determine the minimal fungicidal concentration (MFC), from the concentrations where no growth was observed, including the controls, a subculture was made on SDA and incubated for 24 h at 37 °C. The MFC was defined as the minimum concentration of peptide or extract that eliminated 100% of the growth compared to the control (*n* = 3).

### 2.6. Time–Kill Curve

In honeycomb microplates and using RPMI medium, concentrations of 2 MIC, MIC, 0.5 MIC, and 0.25 MIC of the treatments (150 μL) were evaluated. Then, 150 μL of adjusted inoculum (0.5–2.5 × 10^3^ cells/mL) was added and the plates were incubated in Bioscreen C equipment at 37 °C for 48 h, under constant agitation, and the absorbance reading was performed every hour at 600 nm. The controls used were the same as for the previous procedure. The interpretation was performed as follows: fungistatic: growth inhibition of 50% for 48 h, and fungicide: yeast killing of 99.99% for 72 h (*n* = 3) [57,58].

### 2.7. Checkerboard Assay

This was performed using the checkerboard test, with some adjustments. Briefly, peptide or extract with FLC, peptide with extract, and extract, peptide, and FLC (final concentrations: 0, 0.06, 0.12, 0.25, 0.50, 1, and 2 times the MIC) were mixed. Then, inoculum (final concentration 0.5–2.5×10^3^ cells/mL) was added and incubated at 37 °C for 48 h, after which time the new MIC was established. The fractional inhibitory concentration index (FIC I) was calculated as follows: [A/MIC A] + [P/MIC P] = FIC I, where MIC A and MIC P are the MICs of the individual peptide, extract, and/or antifungal. A and P are the MICs of the same, but in combination [59,60].

The combinations were classified as synergistic (FIC I ≤ 0.5), additive (FIC I > 0.5–1), indifferent (FIC I > 1 < 4), or antagonistic (FIC I > 4).

### 2.8. Cell Viability

Initially, the same procedure described above to determine the MIC and the checkerboard was performed, in the case of the combinations. After 48 h of incubation, the LIVE/DEAD™ Yeast Viability kit from Thermo Fisher Scientific was used and staining was performed according to the supplier’s instructions [61]. Briefly, the multiwell plate was centrifuged for 5 min at 10,000× *g*, the supernatant was discarded, and 80 µL of phosphate buffered saline (PBS), 20 µL of FUN™ 1, and 100 µL of Calcofluor™ White M2R (CW) were added, incubated for 30 min at 30 °C, and placed under an Olympus FV 1000 confocal microscope. Wavelengths used: 405, 532, and 488 nm.

This viability kit combines a two-color fluorescent probe for determining yeast viability, FUN™ 1, and a fluorescent reagent that binds to the cell wall surface, CW. Thus, if the integrity of the plasma membrane is preserved, the metabolic function of the yeast is observed when it converts the yellow-green-fluorescent intracellular staining of FUN™ 1 into red-orange intravacuolar structures. CW labels chitin with blue fluorescence regardless of metabolic status [61].

### 2.9. Scanning Electron Microscopy

The yeast strains (0.5–2.5 × 10^3^ cells/mL) were pretreated with concentrations of 0.5 MIC of peptide, extract, or their combinations for 48 h. Later, the solutions were fixed in this way, a wash was carried out with PBS, and then 100 µL of glutaraldehyde 2.5% was added and left at room temperature for 18 h. The suspension was then washed twice with 70% ethanol at 5 min intervals, three times with 95% ethanol at 10 min intervals, and three times with 100% ethanol at 20 min intervals. Samples were stored in 100% ethanol until being analyzed [18,62]. The subsequent drying and observation of the samples were carried out at Andes University.

### 2.10. Propidium Iodine Staining

The cell membrane permeability generated by the peptide, extract, and combinations was determined by propidium iodide (PI) staining. PI cannot permeate into cells with an intact plasma membrane; however, when cell integrity is compromised, it enters the nucleus, where it complexes with DNA, making the nucleus highly fluorescent [63]. The four study strains (1 × 10^7^ cells/mL) were treated with MIC, sub-MIC of peptide and extract (*C. albicans* SC5314: 25 μg/mL and 250 μg/mL, respectively, *C. albicans* 256: 50 μg/mL and 62.5 μg/mL, respectively, *C. auris* 435: 100 μg/mL and 125 μg/mL, respectively, and *C. auris* 537: 25 μg/mL and 31.25 μg/mL, respectively), and combinations for 4 h at 37 °C. After this time, the cells were washed with PBS and 5 µL of CW (5 µg/mL) and PI (50 µg/mL) were added to the pellet, finishing with PBS to a final volume of 500 µL. Subsequently, the staining was left for 10 min at 4 °C in darkness and washed two times with PBS. The samples were observed under a LEICA DMi8 microscope (Leica Microsystems, Madrid, Spain).

### 2.11. Hemolytic Activity Assay

The hemolytic activity of the ethanolic extract of *B. pilosa*, palindromic peptide, and their combinations was analyzed on human red blood cells [64]. Briefly, 5 mL of peripheral blood from a healthy individual was collected in an EDTA tube and centrifuged at 2500 rpm for 7 min. The erythrocyte-rich fraction was suspended in PBS and washed twice by centrifugation. Then, 100 µL of the *B. pilosa* extracts (15.62–2000 µg/mL), palindromic peptide (1.56–200 µg/mL), or additive or synergistic combinations of peptide and plant extract (combination 1: 25/250 µg/mL, combination 2: 50/62.5 µg/mL, combination 3: 100/125 µg/mL, and combination 4: 25/31.25 µg/mL) was mixed with 100 µL of erythrocytes (2% hematocrit) in round-bottom 96-well plates and incubated at 37 °C for 2 h. Subsequently, the reaction was stopped by centrifuging for 10 min at 2500 rpm. To measure hemoglobin levels, 100 µL of the supernatant containing the hemoglobin released from lysed erythrocytes was transferred to a 96-well plate with a flat bottom, and the absorbance was measured at 450 nm. PBS was used as a negative control, while Tween-20 (20% *v*/*v*) in PBS was used as a positive control (*n* = 3).

### 2.12. Cytotoxicity Assay

The toxicity of the ethanolic extract of *B. pilosa*, the palindromic peptide, and their combinations was evaluated in a primary culture of human foreskin fibroblasts by the MTT (tetrazolium salt, Sigma-Aldrich) assay [48], with modifications. Briefly, cells were seeded in 96-well plates at a density of 1 × 10^4^ cells and 100 μL per well and allowed to adhere and proliferate for 24 h RPMI medium at 37 °C and 5% CO_2_.

Following this, the complete medium was replaced with incomplete medium (without 10% fetal bovine serum) for an additional 24 h in order to achieve synchronization. The cells were then incubated at 37 °C and 5% CO_2_ for 2 h with 100 μL of *B. pilosa* extract (31.25–2000 µg/mL), peptide at the concentrations evaluated (3.12–200 μg/mL), and additive or synergistic combinations (C 1: 25/250 µg/mL, C 2: 50/62.5 µg/mL, C 3: 100/125 µg/mL, and C 4: 25/31.25 µg/mL). The culture medium was removed, and 100 μL of incomplete medium with 10% MTT was added and incubated at 37 °C and 5% CO_2_ for 4 h. Subsequent to the incubation period, the MTT solution was removed, and 100 μL of DMSO was added to dissolve the formazan crystals. Following a 30 min incubation at 37 °C, the absorbance at 575 nm was measured, using the BioTek ELx800 absorbance reader (Agilent, Santa Clara, USA). An incomplete culture medium with 10% MTT was used as a negative control. In addition, the half-maximal inhibitory concentration (IC_50_) was calculated by plotting viability versus log (concentration).

## 3. Results and Discussion

### 3.1. Antifungal Activity of Peptides against C. albicans and C. auris

In this investigation, the antifungal activity of twenty-one peptides derived from the peptide RWQWRWQWR, named R-1-R, was evaluated against the reference strain *C. albicans* SC5314 (sensitive to antifungals) and clinical isolates *C. albicans* 256-PUJ-HUSI (FLC-resistant), *C. auris* 435-PUJ-HUSI (FLC-sensitive), and *C. auris* 537-PUJ-HUSI (FLC- and AmB-resistant) (Table 1). The peptides used in this investigation were synthesized manually via solid-phase peptide synthesis using the Fmoc/tBu strategy, and these molecules were classified as follows:

Group I: sequences in which each amino acid is substituted sequentially by alanine (alanine scan) [48]; these peptides were evaluated in order to determine which amino acids are relevant for antifungal activity.

Group II: Lysine (Lys) was substituted for all arginine (Arg) residues; replacing Arg residues with Lys has been shown to facilitate and reduce the cost of the synthesis [44].

Group III: Some Arg residues were removed or added at the N-terminal and/or C-terminal ends to determine the influence of the positive charge in the antifungal activity [46,65,66].

Group IV: Previous reports have shown that the substitution of Arg by Ala at position 5 increased the cytotoxic effect in breast cancer cell lines. To establish whether this position is relevant in antifungal activity, position 5 was replaced by non-natural hydrophobic amino acids [42,65]; additionally, the presence of non-natural amino acids in the sequence increased the proteolytic stability [67,68,69].

To facilitate reading, all peptides used in this research were coded, taking into account the modification performed on central sequence WQWRWQW, which is coded as 1; hence, palindromic peptide RWQWRWQWR corresponds to R-1-R. For the peptides generated as alanine scan or modifications in sequence 1, the residue position changed is indicated in superscript, and the residue introduced is indicated in square brackets; for example, peptide RWAWRWQWR corresponds to [^3^A]-R-1-R and KWQWKWQWK to K-1-K (Table 1).

The results showed that the palindromic peptide R-1-R had significant antifungal activity against *C. albicans* SC5314 and 256 (MIC and MFC values were 67 µM in all cases). Peptides where substitution of glutamine (Gln) by Ala was performed, [^3^A]-R-1-R and [^7^A]-R-1-R (group I), exhibited similar antifungal activity against the reference strain and greater activity against the clinical isolate than the R-1-R peptide, suggesting that the increased hydrophobicity at position 3 or 7 is relevant for antifungal activity. The Gln side chain does not contribute charge to the sequence and does not influence the amphipathic properties, so it can be considered a spacer. Additionally, peptide [^7^A]-R-1-R was previously studied, and it exhibited increased antibacterial activity against *E. coli* 25922 compared with R-1-R [48]. These results suggest that the absence of Gln at the 7th position does not decrease the antimicrobial activity.

Peptides in which the tryptophan (Trp) or Arg residues were replaced by Ala or peptides in which the net positive charge was reduced exhibited lower antifungal activity against both strains of *C. albicans* than the R-1-R peptide, suggesting that these changes affect the amphipaticity of the sequence, which is relevant to antifungal activity. Furthermore, substitution of any Arg or Trp residue in the sequence decreased the antifungal activity, suggesting that Arg and Trp residues are relevant for antifungal activity.

Substitution of Arg or Trp by Ala in the palindromic sequence affects the net charge and amphipathic properties. Arg and Trp residues have been related to the interaction of the peptide with negatively charged molecules on the cell surface and the lipid bilayer, respectively. These results are in agreement with previous studies performed with Ala scan analogues of the R-1-R peptide against Gram-positive and Gram-negative bacteria and cancer cells [48], where the substitution of Arg or Trp with Ala significantly reduced the biological activity, indicating that the amphipathicity generated by the cationic charge of Arg and the hydrophobicity of Trp are crucial for the antifungal activity against *C. albicans* strains, thus also being consistent with the initial mechanism of antimicrobial action proposed for peptides derived from LfcinB, which is related to electrostatic and hydrophobic interactions with cell membranes [13,19].

Peptides K-1-K, RR-1-R, and R-1-RR exhibited similar antifungal activity against both *C. albicans* strains, indicating that the substitution of Arg by Lys and the incorporation of an Arg residue at the N-terminal do not reduce the antifungal activity. These results suggested that the positive charge could be relevant to antifungal activity. These results are consistent with studies that showed that Arg by Lys substitutions in the AMP sequence induced equal or less antibacterial activity [70,71,72]. The action mechanism of AMPs is based on the initial interaction between the negatively charged molecules on the cell surface and the positively charged side chains of the peptide. The substitution of Arg by Lys does not change the net charge of the sequence; however, the nature of each side chain is different: the primary amine of the guanidine group interacts with the phospholipids of cell membranes and has a more distributed positive charge, so it can form a greater number of hydrogen bonds [70,71,72].

Peptide RR-1-R resulted from the addition of an Arg residue at the N-terminal of the sequence of the palindromic peptide and exhibited similar antifungal activity against *C. albicans* strains. When the Arg residue was incorporated into the N-terminal end, the minimal motif of the LfcinB (RRWQWR) was completed in the peptide sequence. This result is in agreement with previous reports that showed that peptides containing the minimal motif exhibited antibacterial, antifungal, antiviral, and anticancer activity [50]. Peptides RR-1-RR and R-1-RR exhibited antifungal activity only against the SC5314 strain, while the antifungal activity against the isolated clinical *C. albicans* 256 was reduced. Our results suggested that the increases in the net positive charge in the sequence did not increase the antifungal activity.

In peptides from group IV, the central Arg residue was replaced by non-natural hydrophobic amino acid. The antifungal activity against *C. albicans* strains of all peptides was lost, suggesting that the Arg residue in this position is important for antifungal activity. These last results are in agreement with studies that showed that an increase in the positive charge or hydrophobicity did not improve the antibacterial activity of R-1-R; in this case, the antifungal activity was not improved either. The same thing happened in a bioinformatics modeling study with short cationic AMPs, where going from a +4 to a +7 charge also negatively affected the activity [46,66,73]. Our results agree with previous studies [21], and peptide R-1-R had antifungal activity against strains and clinical isolates of *C. albicans*, *C. glabrata*, *C. krusei*, *C. auris*, and *C. tropicalis*. Additionally, it was fungistatic and fungicidal and combined with FLC exhibited a synergistic antifungal effect against *C. tropicalis* 883 and *C. krusei* 6258 (resistant to FLC).

The antifungal activity of peptide R-1-R against *C. albicans* was comparable to that reported for LFB (MIC 200->6,400 μg/mL) and LfcinB (MIC 0.8–400 μg/mL). Additionally, the antifungal activity of the peptide R-1-R is in agreement with that previously reported for short peptides derived from LfcinB against other *C. albicans* strains: Peptide 2 (FKCRRWQWRM; MIC 17.3–17.5 μM), bLF (or LFB) 17–30 (FKCRRWQWRMKKLGA; MIC 5–10 μM), LfcinB-20 (LfcinB 18–37) (FKCRRWQWRMKKLGA; MIC 8 μg/mL), Lfcin B-18-42 (MIC 100 μg/mL); Lfcin B-9 (RRWQWRMKK; MIC 25,000–32,000 μg/mL), Peptide 5 (RWQWRM; MIC 500 μM), and Peptide 3 (GAPSITCVRRAF; MIC 635 μM). The peptide R-1-R also exhibited similar antifungal activity to other palindromic sequences rich in Arg and/or Trp residues in some *C. albicans* strains, as follows: IRIRIRIR, KWKWWKWK, KWKWKWKW [21].

The foregoing supports the idea that not only are a higher cationic charge and hydrophobic residues important for antimicrobial activity but so is a balance between these two parameters, which, although they are important in the mechanism of action in AMPs, are not the only ones. For example, it is reported that the cationic nature and unique geometry of the hydrogen bonds of the Arg residue together with the complex properties of Trp can complement each other to exert an antimicrobial effect. The cationic charge of Arg provides an efficient means of attracting peptides to target membranes, and hydrogen bonds facilitate their interaction with negatively charged surfaces (effectively achieved with other cationic residues), allowing Trp (the most suitable amino acid according to studies) a prolonged association of the peptide with the cell membrane, although this interaction model depends on the AMP [74].

The palindromic peptide R-1-R exhibited low antifungal activity against the two clinical isolates of *C. auris* evaluated, while peptide [^5^A]-R-1-R was the one peptide that exhibited the highest antifungal activity against the clinical isolate *C. auris* 537. Our results suggest that the palindromic peptide exhibited less antifungal activity against *C. auris* strains than those of *C. albicans*. In a previous report, we showed that peptides [^5^A]-R-1-R and R-1-R exhibited similar antibacterial activity against *E. coli* and *S. aureus* strains. Additionally, the peptide [^5^A]-R-1-R exhibited a higher cytotoxic effect against colon cancer cells Caco-2 than peptide R-1-R [48,49].

### 3.2. Antifungal Activity Extracts of B. pilosa against C. albicans and C. auris

#### Phytochemical Analysis

A chemical profile of the *B. pilosa* extract was achieved via UPLC-PAD and mass spectrometry (UHPLC-MS). The results showed the presence of at least eleven compounds (Figure 1 and Appendix A) (Table 2).

Taking into account parameters such as mass and UV–Vis spectra, the assignment of the compounds was made in accordance with the agreement with the information reported in the literature. Peak 1 presented *m*/*z* 353.0857, which was confirmed as chlorogenic acid, including comparison with the reference standard. Other caffeoylquinic acids identified were peaks 4 and 5. Peaks 2, 3, 6, 9, and 10 were identified as glycosylated flavonoids, with compound 10 being observed as the major peak and identified as Okanin 4′-*O*-(triacetyl)-glucoside. Okanins are chalcones commonly found in the genus *Bidens* [75]. Finally, for compounds 7, 8, and 11, no agreement was achieved with what is reported in the literature, and they were labeled as unidentified.

**Table 2 jof-09-00817-t002:** Species identify in the extracts of *B. pilosa*.

No.	Retention Time (min)	λ mx/nm	[M − H]-m/z	Identification	References
1	1.52	244/325	353.0867	Chlorogenic acid *	
2	2.13	248/356	609.1423	Rutin *	
3	10.46	252/368	449.1077	Okanin-*O*-glucoside	[52,76]
4	11.22	242/328	515.1183	3,4-di-Caffeoylquinic acid	[77,78]
5	12.29	242/328	515.1177	3,5-di-Caffeoylquinic acid	[77,78]
6	17.75	236/372	533.1292	Okanin 4′-*O*-(diacetyl)-glucoside	[52]
7	18.81	Not observed	513.1389	Unidentified	
8	19.26	Not observed	409.1480	Unidentified	
9	20.78	240/377	575.1402	Okanin 4′-*O-*(triacetyl)-glucoside	[75,79]
10	21.38	240/377	575.1406	Okanin 4′-*O*-(triacetyl)-glucoside (isomer)	[79]
11	23.51	Not observed	679.1653	Unidentified	

λ mx/nm: maximum absorbance values, [M − H]-*m*/*z*: mass/charge ratio, * compounds identified by reference standard.

The antifungal activity of three *B. pilosa* extracts was initially evaluated against *C. albicans* SC5314 and 256-PUJ-HUSI (FLC-resistant), where the lowest MICs were achieved with the leaf extract obtained by ethanolic maceration (MIC 500 μg/mL and MFC 1000 μg/mL). Subsequently, this same extract was evaluated against *C. auris* 435-PUJ-HUSI and 537-PUJ-HUSI (FLC- and AmB-resistant), achieving similar antifungal activity (MIC 500 μg/mL and MFC 1000->2000 μg/mL) (Table 3). The activity of this extract was greater compared to that reported by Fernando et al., who evaluated the ethanolic extract by disc diffusion, where they achieved inhibition halos for *C. albicans* at extract concentrations between 250–1250 μg/mL and 50 mg/mL [24]. Similarly, Angelini et al. [28] reported that the methanolic extract of *B. pilosa* had MIC values between 125 and 250 μg/mL (as described by CLSI method [54,55,56]) against *C. albicans* YEPGA 6183.

Little is known about the antifungal activity of *B. pilosa* extracts. An in silico study showed that caftaric acid, a predominant phenolic acid in methanolic extracts of the leaves, showed high affinity for lanosterol 14 alpha-dimethylase [22], an enzyme involved in the formation of the yeast cell membrane. Antifungal activity of other phenolic acids, such as chlorogenic acid, has been reported against *Candida* spp. resistant to FLC, generating a death process similar to apoptosis [80]. Caffeic acid and its ester derivatives showed activity against planktonic cells and a biofilm of *C. albicans* [81] and *C. auris* resistant to antifungal [82], and gallic acid is active against *C. albicans, C. glabrata*, and *C. tropicalis*, with MICs between 12.5 and 100 μg/mL [83]. Lastly, activity has also been reported with the flavonoid rutin against *C. albicans* (1000 μg/mL) [84].

The antifungal activity of caffeic acid, caftaric acid, chlorogenic acid, gallic acid, and rutin against *C. albicans* was evaluated (Table 3), showing that only gallic acid (62.5–125 µg/mL, according to the strain evaluated) had a MIC value 4 to 8 times lower than the *B. pilosa* extract, suggesting that the antifungal activity of the extract may be mainly influenced by the presence of this acid. Despite this, the gallic acid content of the extract may be low, the reason for which was not found in the initial phytochemical profile. This suggests that the antifungal activity may be due to various components of the extract, which could act in combination.

### 3.3. Growth Inhibition and Killing Kinetics

Time–kill curves were constructed with the peptide R-1-R and *B. pilosa* extract against the four study strains, using concentrations equivalent to 2 MIC, MIC, 0.5 MIC, and 0.25 MIC values. For R-1-R, the results of the fungistatic and fungicidal activity were previously reported [21]. The palindromic peptide was fungistatic at concentrations of 25 μg/mL (17 μM) against *C. albicans* SC5314 and 256, at 100 μg/mL (135 μM) against *C. auris* 435, and at 400 μg/mL (270 μM) against *C. auris* 537. The peptide was also fungicidal against all strains except *C. auris* 537 (*C. albicans* SC5314: 50 μg/mL (34 μM), *C. albicans* 256: 100 μg/mL (67 μM), *C. auris* 435: 400 μg/mL (270 μM), and *C. auris*: >400 μg/mL (>270 μM)).

Regarding the extract of leaves obtained by ethanolic maceration of *B. pilosa* (Figure 2, Table 4), it exhibited a fungistatic effect for the four strains evaluated: in particular, *C. albicans* SC5314 and *C. auris* 537 at 250 μg/mL and 125 μg/mL, respectively. The extract decreased the growth of the yeasts by ~50% at up to 48 h of incubation and prolonged the adaptation stage up to approximately 10 h. Against *C. albicans* 256, the extract of 500 μg/mL reduced the growth by ~80% after 48 h of incubation, prolonging the adaptation stage up to 35 h. Finally, against *C. auris* 435, the extract induced a decrease in growth of ~50% and ~70% at concentrations of 250 μg/mL and 1000 μg/mL, respectively. The extract of *B. pilosa* exhibited a fungicidal effect for *C. albicans* SC5314, *C. albicans* 256, and *C. auris* 537 (500–1000 μg/mL), but not against *C. auris* 435, at up to 48 h of incubation.

### 3.4. Antifungal Activity of Mixtures of Peptide/Extract, Extract/FLC, and Peptide/FLC

The combination of new antifungal treatment alternatives (AMPs, plant extracts, or synthetic molecules) with commercial antifungal agents has been used to overcome the challenge of antifungal resistance and toxicity [43,60,85,86]. In the present study, after confirming that both the palindromic peptide R-1-R and the ethanolic extract of *B. pilosa* leaves exhibited weak to moderate activity (according to those described by cut-off points suggested by [60]) against two species of *Candida*, both sensitive and resistant to antifungals, they were evaluated in combination with FLC and with each other, to see if this antifungal activity could be improved.

Initially, the antifungal activity of mixtures of R-1-R/FLC against the *C. albicans* and *C. auris* strains was evaluated (Table 5). This mixture exhibited an additive effect against *C. albicans* SC5314 (sensitive to FLC) and *C. albicans* 256 (resistant to FLC) with FIC Index (FIC I) of 0.75 and 0.62, respectively. The MIC value of FLC in *C. albicans* SC5314 was reduced by a factor of 2, while that in *C. albicans* 256 was reduced by a factor of 8. Interestingly, it was found that *C. albicans* 256 returned to a sensitive phenotype when FLC (4 µg/mL) was combined with R-1-R. In the case of *C. auris* 435 (sensitive to FLC), a synergistic effect was obtained (FIC I = 0.28), enhancing the activity of both the peptide and the FLC, by a factor of 32 and 4, respectively. For *C. auris* 537 (resistant to FLC and AmB), an indifferent effect was obtained (FIC I = 1.1) [21].

Subsequently, the extract of *B. pilosa* was combined with FLC (Table 6), this mixture was evaluated against *C. albicans* SC5314, and additivity was obtained (FIC I = 0.53), decreasing the MIC of the extract and FLC by factors of 32 and 2, respectively. Against *C. albicans* 256 (FIC I = 1.25) and *C. auris* 435 (FIC I = 1.03) and 537 (FIC I = 1.25), an indifferent effect was determined.

When the peptide and the extract were combined, it was possible to obtain the greatest antifungal activity. Since a synergistic effect was found for *C. albicans* SC5314 (FIC I = 0.5), *C. auris* 435 (FIC I = 0.37), and *C. auris* 537 (FIC I = 0.12), with this combination it can be seen that the activity of both the peptide and the extract was enhanced in all strains. For *C. albicans* 256, there was an additivity effect (FIC I = 0.62); however, the MIC of the R-1-R and extract was reduced by factors of between 2 and 8 (Table 7).

Finally, it was decided to evaluate the combination of extract, peptide, and FLC (Table 8), obtaining additivity against *C. albicans* SC5314 (FIC I = 0.68), *C. albicans* 256 (FIC I = 0.65), and *C. auris* 435 (FIC I = 0.81) and a synergistic effect against *C. auris* 537 (FIC I = 0.12). It is important to point out that *C. albicans* 256 and *C. auris* 537, with this combination, returned to a phenotype sensitive to FLC (MIC 4 and 8, respectively). On the other hand, it was also found that the combination of extract, peptide, and FLC against *C. albicans* 5314 (E: 31.25 µg/mL, P: 50 µg/mL, FLC: 0.06) and 256 (E: 31.25 µg/mL, P: 50 µg/mL, FLC: 4) resulted in a fungicidal effect, unlike all the other combinations evaluated, where the effect was fungistatic.

MIC_E_, MIC_R-1-R_, and MIC_FLC_ correspond to the MIC (µg/mL) of the extract of *B. pilosa*, R-1-R, and FLC, respectively, and E, P, and FLC are the MIC values when combining the extract, peptide, and FLC. Minimum fractional concentration index (FIC I), MIC_E_/E, MIC_R-1-R_/R-1-R, and MIC_FLC_/FLC MIC represent the factor by which the extract, peptide, or FLC are potentiated after being evaluated in combination, respectively.

After the evaluation of the different combinations of extract, peptide, and/or FLC, several important findings were determined. First, the palindromic peptide is key to enhancing the activity of FLC, especially in resistant Candida strains. Previously, R-1-R was evaluated in combination with FLC or caspofungin (CAS) in different Candida species resistant to these antifungals, such as *C. albicans* 256, C. tropicalis 883, C. krusei 6258, *C. auris* 537, and C. glabrata 1875, obtaining a synergistic or additive effect [21]. In the present study, the FLC-sensitive strains *C. albicans* SC5314 and *C. auris* 435 were included. According to the FIC I obtained, it can be concluded that the resistant strains were more susceptible to this combination (Table 5), and the same occurred when the peptide was mixed with the extract and FLC; even the two resistant strains returned to an FLC-sensitive phenotype (Table 8). These results are in agreement with previous studies carried out with LF, LfcinB, and a peptide derived from human LF (hLF) (1–11) [20,87].

Although the mechanism of this synergistic effect is not clear, it has been suggested after studies with the peptide hLF(1–11) [87] that this peptide in subcandidacidal concentrations enters the yeast through interaction with surface molecules of the fungus and goes to the mitochondria, producing extracellular adenosine triphosphate (ATP) release (it has been seen that this released ATP could generate pores in the cell or the mitochondrial membrane [88]), but this is not enough to generate cell death, for which later, when FLC is added (in subcandidacidal concentrations), it acts as an effector and finally induces cell death. This could be extrapolated to LfcinB peptides, since it has also been shown that LfcinB15 generates extracellular ATP release [19], although more studies are needed to confirm whether, for this AMP, ATP can generate pores, release ions, and induce cell lysis.

On the other hand, it has also been suggested that strains resistant to FLC are more susceptible to LF or LfcinB, possibly due to the absence of cytochrome P-450 (important in the biosynthesis of ergosterol in the yeast membrane), which could produce disturbances in the membrane due to alterations in the sterols [20,89]. This makes resistant strains more susceptible to the action of LF, LfcinB, or derived peptides, which, as mentioned before, initially generates membrane damage. Additionally, it has been proposed that LfcinB could inhibit efflux pumps [20,90] (another mechanism of resistance to azoles), which could explain, in this study, the return of strains resistant to FLC to sensitivity, but complementary studies are needed to confirm it.

Second, when the extract combined with FLC was evaluated (Table 6), an indifferent effect was observed in three strains and only synergy was observed for *C. albicans* SC5314, suggesting that this extract is not of great importance for enhancing FLC activity in all strains. However, when the extract was combined with peptide and FLC (Table 8), it was observed that *C. auris* 537 returned to FLC sensitivity, an effect that was not found only by combining R-1-R and FLC.

When the extract was combined with the peptide (Table 7), it was possible to find the lowest FIC I (0.12–0.62) of all the combinations evaluated against the four study strains. It should even be noted that *C. auris* 537, resistant to FLC and AmB, was the most susceptible strain to this mixture, where both the peptide and extract activity were enhanced by a factor of 16. This indicates that in some way the extract is contributing to the mechanism of action of the peptide, but since there is little information on the mechanism of action of *B. pilosa* extracts, little can be inferred about how the extract acts, with FLC or with the peptide. According to what was described by Angelini [28], it is possible that the extract is helping the peptide in the membrane disruption process, but more studies are needed in this regard. This was corroborated by combining the extract with peptides derived from R-1-R: two ([^3^A]-R-1-R, RR-1-R) presented MICs similar to the palindromic peptide and two peptides ([^2^A]-R-1-R, [^5^Bpa]-R-1-R) that did not exhibit activity. The results showed that with the combination peptide/*B. pilosa*, a synergistic effect is achieved in all cases, except for [^3^A]-R-1-R, which was additive, enhancing the activity of the peptides and extract by a factor of between 2 and 16 (Appendix A).

Third, according to the results obtained, it is considered that the combinations of peptide/FLC, peptide/extract, and peptide and extract/FLC, depending on the yeast, could be a promising strategy for enhancing the activity against *Candida* and could be considered for developing new treatments to mitigate the problem of resistance to antifungals.

### 3.5. Microscopy

To visually capture the antifungal effect obtained by the peptide, the extract, and the combination with the greatest activity in the four study strains, confocal laser scanning microscopy (CLSM) (LIVE/DEAD Yeast commercial kit), and scanning electron microscopy (SEM) were performed.

The effect after 48 h of incubation with R-1-R, extract, or the mixture R-1-R/extract (at MICs and sub-MICs where there was a synergistic or additive effect) was evaluated for the four strains using the LIVE/DEAD™ Yeast Viability kit (Appendix A). Live control without treatment was used with a control of hypochlorite-treated dead yeast cells (results not shown). Live control cells were labeled with a blue fluorescence (Calcofluor White M2R), highlighting cell wall chitin regardless of metabolic state; yeasts were also labeled with FUN1 using two different excitation filters (488 and 532 nm), revealing yeast with a diffusely distributed green and red color, indicating plasma membrane integrity and the yeast cells’ metabolic capability to convert this intracellular staining to a red-orange fluorescence.

For *C. albicans* SC5314 (Appendix A), it was observed that although the metabolic viability was not affected, the number of cells was reduced compared to the live control. For example, with the peptide at 100 µg/mL, no presence of yeasts was observed. This concentration corresponds to the value of the MFC according to the technique of microdilution in broth (Table 1). With the extract at 500 µg/mL (MIC), a decrease of more than ~50% in the cell population was seen. The peptide at 25 µg/mL (0.25 MIC) and the extract at 250 µg/mL (0.5 MIC) (concentrations where there was a synergic effect) also generated a reduction in the number of cells, but less than ~50%. Finally, when peptide (0.25 MIC) and extract (0.5 MIC) were evaluated in combination, it was observed that there was a reduction of approximately ~90% in the cell growth.

A similar effect was seen in the other study strains (Appendix A), with the difference that for *C. albicans* 256 and *C. auris* 435, in the case of combinations of peptide and extract, the number of cells was not reduced by more than ~50%, but rather the metabolic viability; this is inferred from the decreased red color staining in FUN1.

After exposing the yeasts for 48 h to 0.5 MIC, concentrations of peptide and extracts alone and in combination, by SEM, it was observed that compared to untreated yeast in *C. albicans* SC5314 (Figure 3a), the peptide at 50 µg/mL generates an alteration in the cell wall, forming protuberances in the cell in the form of yeast and indentations in the hyphae. On increasing the resolution of the image (500 nm), it was seen as a veil on the cell surface, which could correspond to leakage of the cytoplasmic content. With the extract at 250 µg/mL, it was observed that yeast has a more rounded appearance and there are cracks in the cell wall and blisters in the hyphae. In the case of the combination, a greater disturbance in the cell surface and inhibition of filamentation were observed.

For *C. albicans* 256 (Figure 3b), it was observed that there is also a disturbance in the membrane with peptide (50 µg/mL) and extract (250 µg/mL) alone; specifically, blisters were observed on the surface of the yeast, but in the combination (P: 50 µg/mL and E: 62.5 µg/mL), where there was an additive effect, the damage was more significant, with a rupture of the cell wall being observed. Another important aspect to point out is that compared to FLC-sensitive *C. albicans* SC5314, for *C. albicans* 256, which is resistant to FLC, the presence of hyphae was not observed in the untreated control.

When *C. auris* 435 (Figure 4a) and 537 (Figure 4b) were treated with peptide (100 µg/mL) and extract (250 µg/mL) alone, damage to the cell wall with indentations and protuberances was also observed. With the combinations of peptide and extract there was rupture of the cell wall, and in *C. auris* 537, even leakage of intracellular content is seen.

The results obtained are consistent with previous studies, carried out with peptides derived from LfcinB [18,19], where damage was also seen on the cell surface, and they are also consistent with the mechanism of action proposed for this AMP. Regarding the extract, this is the first study where SEM was performed and damage to the surface of the yeast was also observed. This could contribute to the results obtained in silico, where it has been proposed that the mechanism of action for extracts of *B. pilosa* is directed to the cell membrane [28].

To corroborate that the cell surface damage seen in SEM was related to the increased permeability in the cell membrane, the PI assay was performed at the value of the MICs, subinhibitory concentrations, and combinations of peptide and extract, where the synergistic or additive effect against the four study strains was seen (Figure 5, Figure 6, Figure 7 and Figure 8). After 4 h of incubation and in the case of peptide, combinations, and, to a lesser extent, for the extract (according to the strain), the presence of positive PI cells was found (red fluorescence) compared to untreated controls, highlighting a greater effect of permeability of the combinations in strains resistant to FLC.

The hemolytic activity of R-1-R, *B. Pilosa* leaf extract, and their combinations was assessed against human erythrocytes in order to determine their effect on erythrocyte membrane integrity. The results showed that peptide and extract alone are safe and non-toxic at a wide range of concentrations, including the MIC and MFC, 1.56–200 and 15.62–2000 μg/mL, respectively. At high concentrations of 200 (peptide) and 2000 μg/mL (extract), a slight hemolytic effect on human red blood cells was observed, with percentages of 8% and 4%, respectively (Figure 9); however, they are not considered hemolytic, because they have percentages of less than 10% [91].

When R-1-R and *B. pilosa* leaf extract were evaluated in combination at concentrations where there was synergy or additivity, they did not exhibit significant hemolytic activity (0–2%); on the contrary, it was observed that all combinations decreased the hemolysis percentage by a factor of 2 in comparison with the peptide or the extract alone, while the positive control Tween-20 completely lysed the red blood cells.

In vitro toxicity testing can reveal the risk that may be associated with the use of AMPs and metabolites from plant extracts. In previous studies, Lys/Arg substitutions decreased the hemolytic activity of peptide R-1-R, preserving the antibacterial activity [70]. In the present study, it was demonstrated that the combination of R-1-R and the extract of *B. pilosa* is also an option for reducing the percentage of hemolysis, but it also increases the antifungal activity. This is mainly due to the fact that a greater antifungal activity is achieved with lower concentrations of peptide and extract.

On the other hand, many plants contain chemical substances that might have a hemolytic effect on erythrocytes. The ability of the plant extracts to lyse red blood cells could be dependent on different tannins and saponins that can cause cell disruption [92,93]; however, neither of these two metabolites was found in the extract from *B. pilosa* leaves. In contrast to the metabolites found (Table 2), it has not been found to exhibit considerable hemolytic activity at high concentrations [78,94,95]. Additionally, a study showed that both ethanol and ethylacetate/ethanol extracts from *B. pilosa* plants had a protective effect in normal human erythrocytes against oxidative hemolysis in vitro [96].

Peptide, extract, and their synergistic combinations were subjected to cytotoxicity analysis on a primary cell culture of fibroblasts. The IC_50_ and R^2^ values were 175.9 μg/mL and 0.69 μg/mL, respectively, for R-1-R, while for *B. pilosa* leaf extracts, the IC_50_ and R^2^ values were 657.6 μg/mL and 0.96 μg/mL, respectively. The peptide showed no appreciable toxicity against fibroblasts at 1.56–150 μg/mL, with percentages of cell viability above 50%. However, the extract at concentrations above 1000 μg/mL decreased the percentage of cell viability by more than 50% (Figure 10). Interestingly, when they were evaluated in combination concentrations where there was synergy or additivity, no toxic effect was observed at the tested concentrations of peptide and extract (C1: 25/250 µg/mL, C2: 50/62.5 µg/mL, C3: 100/125 µg/mL, and C4: 25/31.25 µg/mL), with percentages of cell viability above 80%.

Similarly, Barragán-Cárdenas et al. have reported that this palindromic peptide exhibits a rapid cytotoxic effect on breast cancer cells. They also tested its selectivity on primary fibroblasts, non-cancerous MCF-12 cells, and BMEC bovine mammary cells, with minimal effect on them [47,48].

Phytochemical analysis has revealed that *B. pilosa* leaf extracts contain a broad chemical constitution, suggesting that the cytotoxicity activity could be related to secondary metabolites such as terpenes, flavonoids, and alkaloids [97]. In previous studies, it was suggested that the strong in vitro activity at high concentrations is related to other mechanisms, such as interference with mitochondrial or lysosomal function [98].

However, the combination did not induce any adverse effects on the fibroblast cells and erythrocytes. Several studies have demonstrated that the combination of drugs offers numerous advantages over monotherapy, which include reduced dosage, synergistic response, decreased possibility of resistance development, and reduced toxicity [99]. Combinations of antimicrobials and NPs that exhibit synergism will possibly show effective pharmacological results [49].

It is possible that secondary metabolites such as hydroxycinnamic acid (HCA) derivatives such as caffeoylquinic acids, flavonoid glycosides, and the di-caffeoylquinic acids and their 3,4- and 3,5-geometric isomers that have been found in *B. pilosa* may improve or facilitate the interaction with the peptide. This strategy has helped prevent the development of resistance while also minimizing toxicity, by allowing the use of lower concentrations of both agents [100]. In order to comprehensively comprehend the potential health advantages and associated risks related to their consumption, additional research is required.

## 4. Conclusions

The results obtained in the present study suggest that the antifungal activity of R-1-R peptide could be dependent on the amphipathicity, and in the case of the *B. pilosa* extract, it is suggested that the activity could be related to a synergistic effect between phenolic acids and the glycosylated flavonoids. The combination between the R-1-R peptide and the ethanolic extract of *B. pilosa* leaves is a strategy for enhancing the antifungal activity against *C. albicans* and *C. auris* sensitive and resistant to antifungals, decreasing their cytotoxicity in human erythrocytes and fibroblasts. Additionally, the combination of peptide/FLC or peptide/extract/FLC managed to return *C. albicans* 256 and *C. auris* 537 from a FLC-resistant to FLC-sensitive phenotype, so these mixtures are seen to be a novel strategy for combating antifungal resistance. Therefore, the combination of NPs and commercial antifungal agents is considered to be a promising therapeutic alternative for the treatment of diseases caused by *Candida* spp.

## Figures and Tables

**Figure 1 jof-09-00817-f001:**
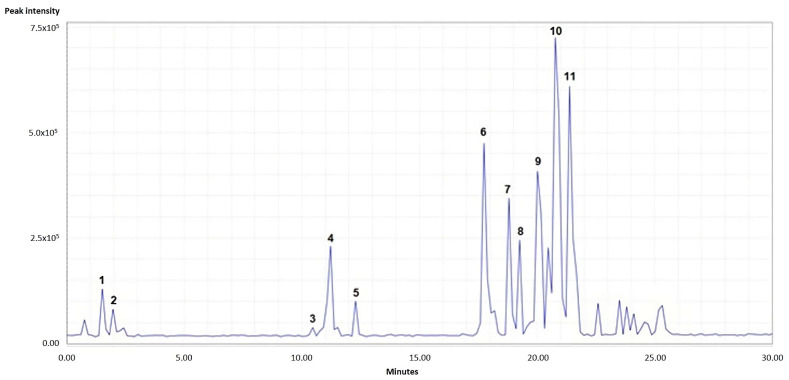
UHPLC-ESI-MS qToF chromatogram of the *B*. *pilosa* leaf ethanolic extract in negative ion mode.

**Figure 2 jof-09-00817-f002:**
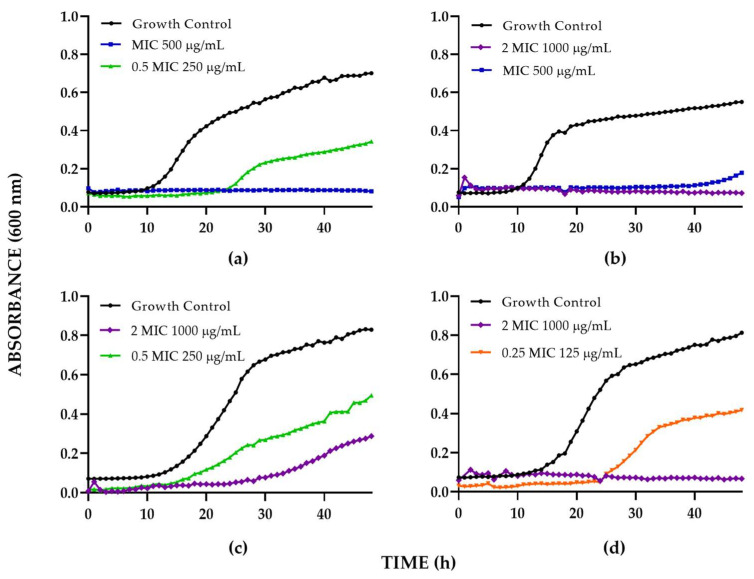
Time–kill curve of *B. pilosa* extract against (**a**) *C. albicans* SC5314; (**b**) *C. albicans* 256; (**c**) *C. auris* 435; (**d**) *C. auris* 537.

**Figure 3 jof-09-00817-f003:**
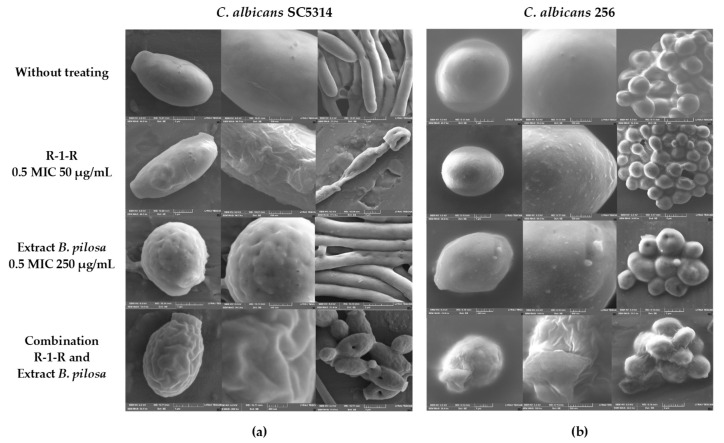
SEM of (**a**) *C. albicans* SC5314 and (**b**) *C. albicans* 256 after treatment (48 h) with R-1-R, *B. pilosa* extract to 0.5 MIC values, and combinations of peptide and extract where there was synergistic effect. R-1-R (25 µg/mL) and *B. pilosa* extract (250 µg/mL) to *C. albicans* SC5314 and R-1-R (50 µg/mL) and *B. pilosa* extract (62.5 µg/mL) to *C. albicans* 256. Image size: Column one: 1 µm and column two: 500 nm.

**Figure 4 jof-09-00817-f004:**
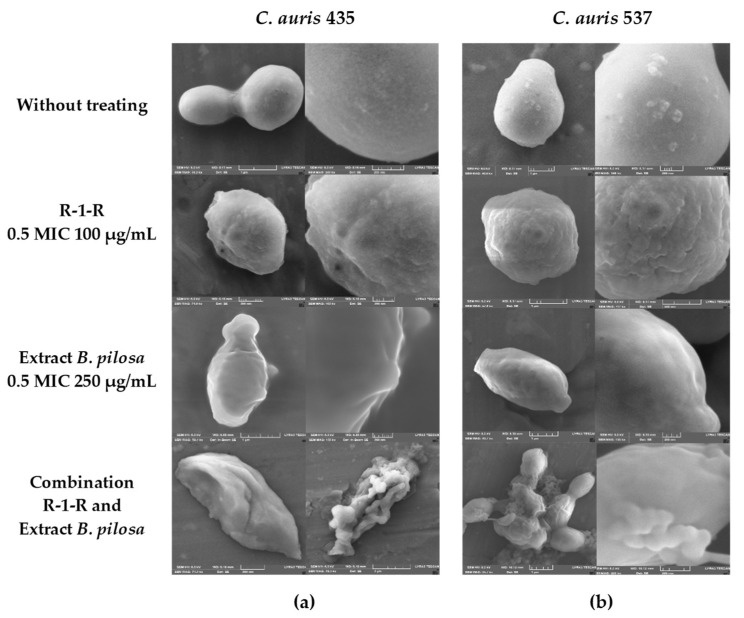
SEM of (**a**) *C. auris* 435 and (**b**) *C. auris* 537 after treatment (48 h) with R-1-R, *B. pilosa* extract to 0.5 MIC values, and combinations of peptide and extract where there was synergistic effect: R-1-R (100 µg/mL) and *B. pilosa* extract (125 µg/mL) to *C. auris* 435 and R-1-R (25 µg/mL) and *B. pilosa* extract (31.25 µg/mL) to *C. auris* 537. Image size: Column one: 1 µm and column two: 500 nm.

**Figure 5 jof-09-00817-f005:**
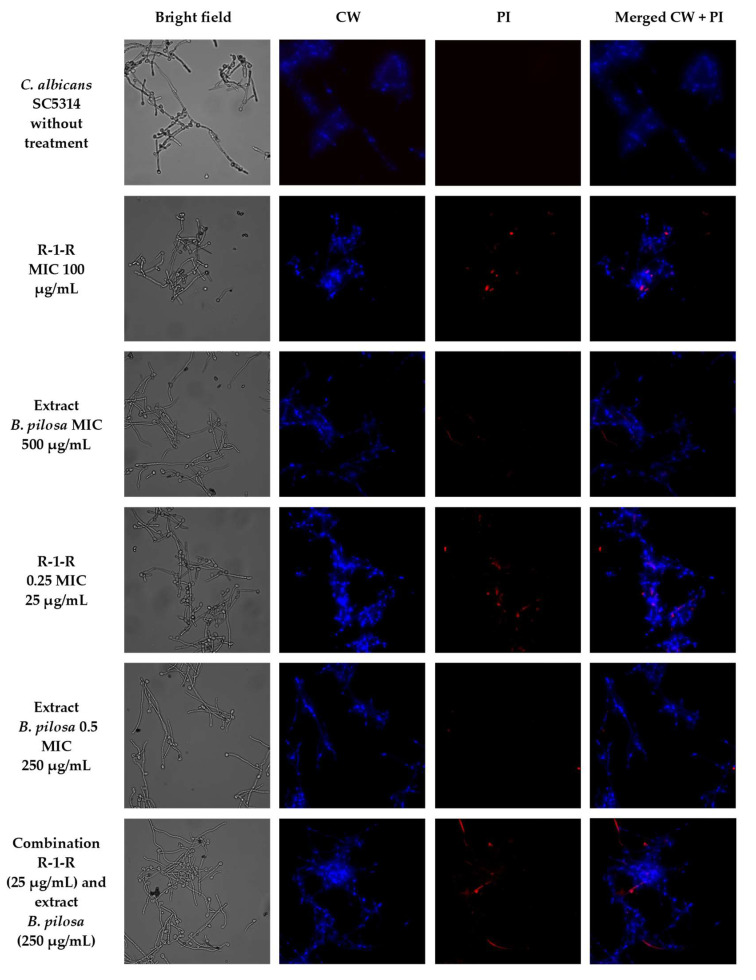
Propidium iodide staining of *C. albicans* SC5314 after treatment (4 h) with R-1-R, *B. pilosa* extract at MIC, sub-MIC values and combinations of peptide and extract where there was a synergistic effect.

**Figure 6 jof-09-00817-f006:**
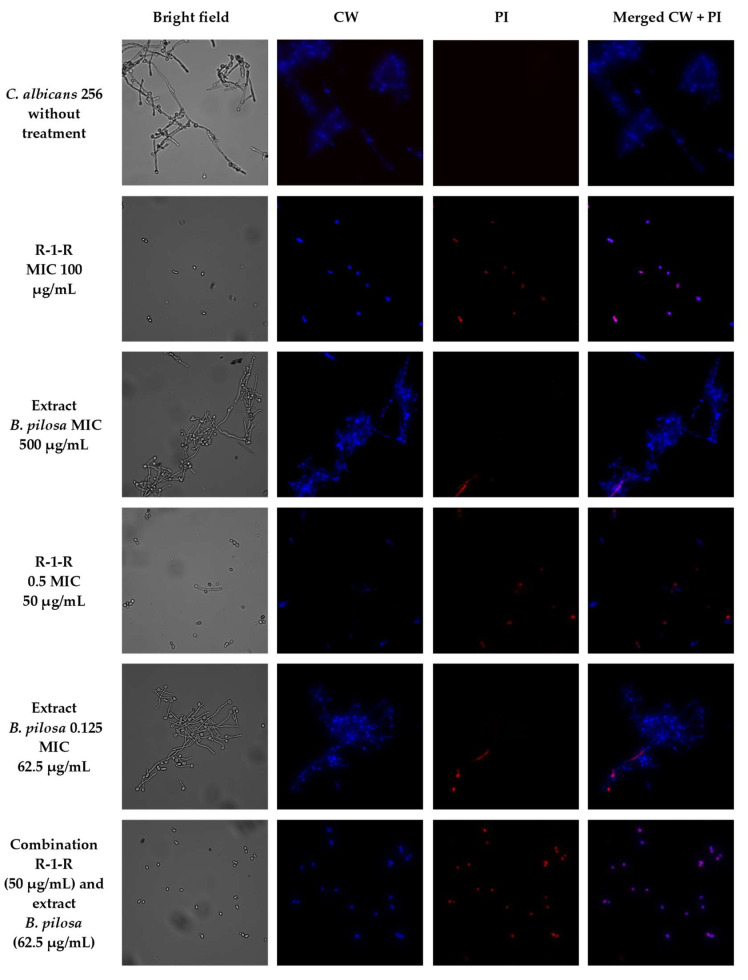
Propidium iodide staining of *C. albicans* 256 after treatment (4 h) with R-1-R, *B. pilosa* extract at MIC, sub-MIC values and combinations of peptide and extract where there was an additive effect.

**Figure 7 jof-09-00817-f007:**
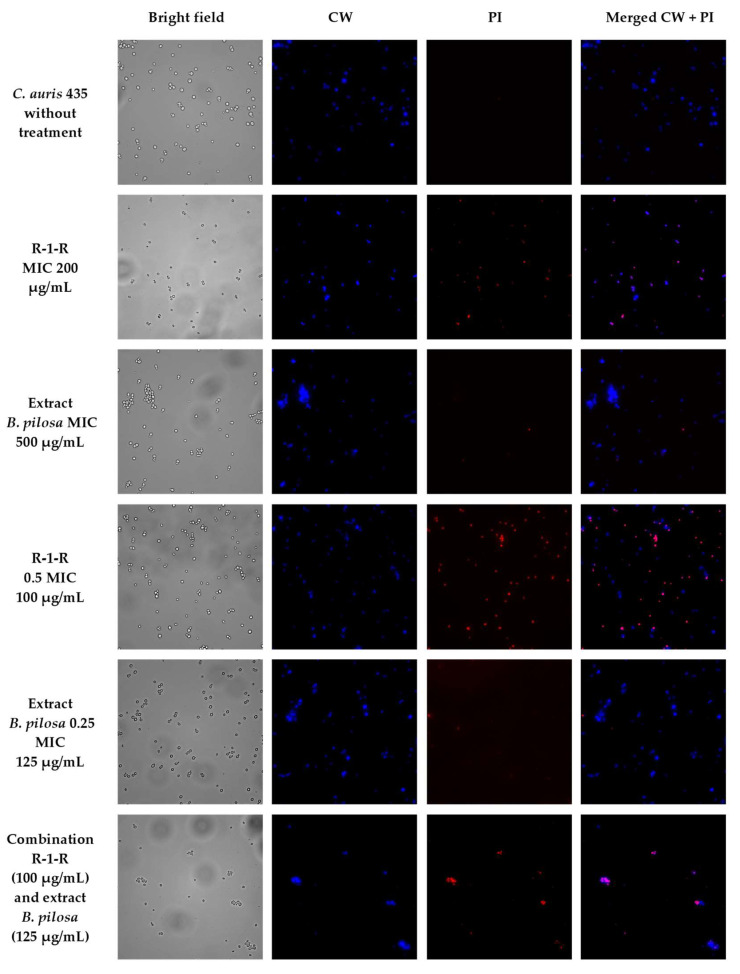
Propidium iodide staining of *C. auris* 435 after treatment (4 h) with R-1-R, *B. pilosa* extract at MIC, sub-MIC values and combinations of peptide and extract where there was a synergistic effect.

**Figure 8 jof-09-00817-f008:**
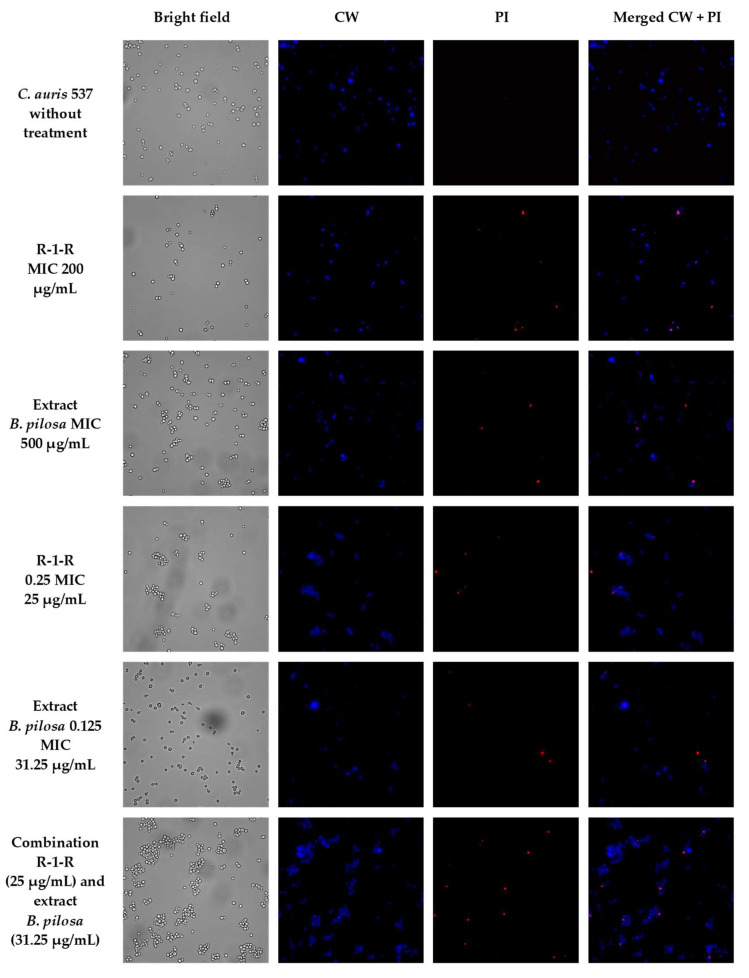
Propidium iodide staining of *C. auris* 537 after treatment (4 h) with R-1-R, *B. pilosa* extract at MIC, sub-MIC values and combinations of peptide and extract where there was a synergistic effect.

**Figure 9 jof-09-00817-f009:**
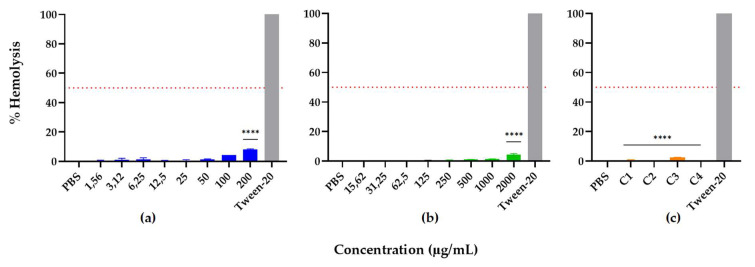
Hemolytic activity of (**a**) R-1-R, (**b**) extract of *B. pilosa*, and (**c**) combinations of peptide and extract where there was synergistic or additive effect, C1: 25/250 µg/mL, C2: 50/62.5 µg/mL, C3: 100/125 µg/mL, and C4: 25/31.25 µg/mL. Dotted red line indicates 50% hemolysis. *p*-values of <0.05 were used to indicate statistical significance as follows: **** *p* < 0.0001, compared to Tween-20 control (*n* = 3).

**Figure 10 jof-09-00817-f010:**
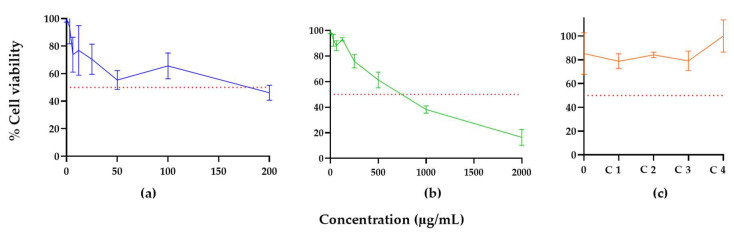
Selective cytotoxic effect of (**a**) R-1-R, (**b**) extract of *B. pilosa*, and (**c**) combinations of peptide and extract where there was synergistic or additive effect, C1: 25/250 µg/mL, C2: 50/62.5 µg/mL, C3: 100/125 µg/mL, and C4: 25/31.25 µg/mL against primary cell culture of fibroblasts. The dotted red line indicates 50% cell viability (*n* = 3).

**Table 1 jof-09-00817-t001:** Antifungal activity of RWQWRWQWR and its analogues against *C. albicans* and *C. auris*, μg/mL (μM).

	Peptide Code	Sequence	*C. albicans*				*C. auris*			
			SC5314		256-PUJ-HUSI		435-PUJ-HUSI		537-PUJ-HUSI	
			MIC	MFC	MIC	MFC	MIC	MFC	MIC	MFC
LfcinB (21–25)_Pal_	R-1-R	RWQWRWQWR	100 (67)	100 (67)	100 (67)	100 (67)	200 (135)	200 (135)	200 (135)	>200 (>135)
Group I	A-1-R	AWQWRWQWR	200 (143)	200 (143)	100 (71)	200 (143)	>200 (>143)	>200 (>143)	>200 (>143)	>200 (>143)
	[^2^A]-R-1-R	RAQWRWQWR	>200 (>146)	>200 (>146)	>200 (>146)	>200 (>146)	>200 (>146)	>200 (>146)	>200 (>146)	>200 (>146)
	[^3^A]-R-1-R	RWAWRWQWR	100 (70)	100 (70)	50 (35)	100 (70)	>200 (>140)	>200 (>140)	>200 (>140)	>200 (>140)
	[^4^A]-R-1-R	RWQARWQWR	>200 (>146)	>200 (>146)	200 (146)	>200 (>146)	>200 (>146)	>200 (>146)	>200 (>146)	>200 (>146)
	[^5^A]-R-1-R	RWQWAWQWR	>200 (>143)	>200 (>143)	>200 (>143)	>200 (>143)	>200 (>143)	>200 (>143)	100 (71)	100 (71)
	[^6^A]-R-1-R	RWQWRAQWR	>200 (>146)	>200 (>146)	200 (146)	>200 (>146)	>200 (>146)	>200 (>146)	>200 (>146)	>200 (>146)
	[^7^A]-R-1-R	RWQWRWAWR	100 (70)	100 (70)	50 (35)	100 (70)	>200 (>140)	>200 (>140)	200 (140)	>200 (>140)
	[^8^A]-R-1-R	RWQWRWQAR	>200 (>146)	>200 (>146)	200 (146)	>200 (>146)	>200 (>146)	>200 (>146)	>200 (>146)	>200 (>146)
	R-1-A	RWQWRWQWA	>200 (>143)	>200 (>143)	>200 (>143)	>200 (>143)	>200 (>143)	>200 (>143)	>200 (>143)	>200 (>143)
Group II	K-1-K	KWQWKWQWK	100 (72)	100 (72)	100 (72)	100 (72)	>200 (>144)	>200 (>144)	>200 (>144)	>200 (>144)
Group III	RR-1-R	RRWQWRWQWR	100 (61)	100 (61)	100 (61)	100 (61)	200 (122)	>200 (>122)	200 (122)	>200 (>122)
	1-R	WQWRWQWR	200 (150)	200 (150)	200 (150)	200 (150)	>200 (>150)	>200 (>150)	>200 (>150)	>200 (>150)
	RR-1-RR	RRWQWRWQWRR	100 (56)	100 (56)	200 (111)	200 (111)	200 (111)	>200 (>111)	200 (111)	>200 (>111)
	R-1-RR	RWQWRWQWRR	100 (61)	100 (61)	100 (61)	200 (122)	200 (122)	>200 (>122)	>200 (>122)	>200 (>122)
	R-1	RWQWRWQW	200 (150)	200 (150)	200 (150)	200 (150)	>200 (>150)	>200 (>150)	>200 (>150)	>200 (>150)
	1	WQWRWQW	200 (170)	200 (170)	200 (170)	200 (170)	>200 (>170)	>200 (>170)	>200 (>170)	>200 (>170)
Group IV	[^5^Bpa]-R-1-R	RWQWBpaQWR	>200 (>132)	>200 (>132)	>200 (>132)	>200 (>132)	>200 (>132)	>200 (>132)	>200 (>132)	>200 (>132)
	[^5^Dip]-R-1-R	RWQWDipQWR	>200 (>143)	>200 (>143)	>200 (>143)	>200 (>143)	>200 (>143)	>200 (>143)	>200 (>143)	>200 (>143)
	[^5^Nal]-R-1-R	RWQWNalQWR	>200 (>131)	>200 (>131)	ND	ND	>200 (>131)	>200 (>131)	ND	ND
	[^5^HomoF]-R-1-R	RWQWHomoFQWR	200 (134)	>200 (>134)	200 (134)	>200 (>134)	>200 (>134)	>200 (>134)	>200 (>134)	>200 (>134)

Bpa: 4-Benzoyl-L-phenylalanine. Nal: Naphthylalanine. Dip: Diphenylalanine. HomoF: Homophenylalanine. ND: Not determined.

**Table 3 jof-09-00817-t003:** Antifungal activity of vegetal extracts of *B. pilosa* against *C. albicans* and *C. auris*. The site of collection of plant material was Mocoa, Department of Putumayo, Colombia.

Organ	Type of Extraction	*C. albicans*		*C. auris*	
		SC5314	256	435	537
		MIC/MFC (μg/mL)			
Leaves	Percolation EtOH 96%	1000/2000	1000/2000	1000/2000	1000/2000
Stem	Percolation EtOH 96%	>2000/>2000	>2000/>2000	>2000/>2000	>2000/>2000
Leaves	Maceration EtOH 96%	500/1000	500/1000	500/2000	500/>2000
	Caffeic acid	>2000/>2000	>2000/>2000	>2000/>2000	>2000/>2000
	Caftaric acid	>1000/>1000	>1000/>1000	>1000/>1000	>1000/>1000
	Chlorogenic acid	>1000/>1000	>1000/>1000	>1000/>1000	>1000/>1000
	Gallic acid	125/250	62.5/125	125/1000	62.5/1000
	Rutin	>2000/>2000	>2000/>2000	>2000/>2000	>2000/>2000

**Table 4 jof-09-00817-t004:** Fungistatic and fungicidal effect of *B. pilosa* extract against *C. albicans* and *C. auris*.

Strain	Effect (μg/mL)	
	Fungistatic *	Fungicide *
*C. albicans* SC5314	250	500
*C. albicans* 256	500	1000
*C. auris* 435	250	>1000
*C. auris* 537	125	1000

* The values correspond to the fungistatic and fungicidal effect after 48 h of incubation.

**Table 5 jof-09-00817-t005:** Synergy test. Combination of R-1-R with FLC against *C. albicans* and *C. auris*.

Strain	MIC_R-1-R_	MIC_FLC_	R-1-R	FLC	FIC Index	MIC_R-1-R_/R-1-R	MIC_FLC_/FLC
*C. albicans* SC5314	200	0.125	50	0.0625	0.75	4	2
*C. albicans* 256 *	100	32	50	4	0.62	2	8
*C. auris* 435	400	8	12.5	2	0.28	32	4
*C. auris* 537 *	200	64	200	8	1.1	1	8

* Previously published in [21]. MIC_R-1-R_ and MIC_FLC_ correspond to the MIC (µg/mL) of the R-1-R and FLC, respectively, and R-1-R and FLC are the MIC values when combining the R-1-R and FLC. Minimum fractional concentration index (FIC I), MIC_R-1-R_/R-1-R, and MIC_FLC_/FLC represent the factor by which the peptide or FLC are potentiated after being evaluated in combination, respectively.

**Table 6 jof-09-00817-t006:** Synergy test. Combination of *B. pilosa* extract with FLC against *C. albicans* and *C. auris*.

Strain	MIC_E_	MIC_FLC_	E	FLC	FIC Index	MIC_E_/E	MIC_FLC_/FLC
*C. albicans* SC5314	1000	0.25	31.25	0.125	0.53	32	2
*C. albicans* 256	500	128	500	32	1.25	1	4
*C. auris* 435	1000	8	31.25	8	1.03	32	1
*C. auris* 537	500	128	500	32	1.25	1	4

MIC_E_ and MIC_FLC_ correspond to the MIC (µg/mL) of the extract of *B. pilosa* and FLC, respectively, and E and FLC are the MIC values when combining the peptides and FLC. Minimum fractional concentration index (FIC I), MIC_E_/E, and MIC_FLC_/FLC represent the factor by which the extract or FLC are potentiated after being evaluated in combination, respectively.

**Table 7 jof-09-00817-t007:** Synergy test. Combination of *B. pilosa* extract with R-1-R against *C. albicans* and *C. auris*.

Strain	MIC_E_	MIC_R-1-R_	E	R-1-R	FIC Index	MIC_E_/E	MIC_R-1-R_/R-1-R
*C. albicans* SC5314	1000	100	250	25	0.5	4	4
*C. albicans* 256	500	100	62.5	50	0.62	8	2
*C. auris* 435	1000	400	125	100	0.37	8	4
*C. auris* 537	500	400	31.25	25	0.12	16	16

MIC_E_ and MIC_R-1-R_ correspond to the MIC (µg/mL) of the extract of *B. pilosa* and R-1-R, respectively; E and R-1-R correspond to the MIC values when combining the extract and peptide. Minimum fractional concentration index (FIC I), MIC_E_/E, and MIC_R-1-R_/R-1-R represent the factor by which the extract or peptide are potentiated after being evaluated in combination, respectively.

**Table 8 jof-09-00817-t008:** Synergy test. Combination of *B. pilosa* extract, R-1-R, and fluconazole against *C. albicans* and *C. auris*.

Strain	MIC_E_	MIC_R-1-R_	MIC_FLC_	E	R-1-R	FLC	FIC Index	MIC_E_/E	MIC_R-1-R_ / R-1-R	MIC_FLC_/FLC
*C. albicans* SC5314	500	100	0.5	31.25	50	0.06	0.68	16	2	8
*C. albicans* 256	1000	100	32	31.25	50	4	0.65	32	2	8
*C. auris* 435	1000	400	4	62.5	200	1	0.81	16	2	4
*C. auris* 537	1000	400	128	31.25	25	8	0.15	32	16	16

## Data Availability

Not applicable.

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
