# Peer review of "Combining the Peptide RWQWRWQWR and an Ethanolic Extract of Bidens pilosa Enhances the Activity against Sensitive and Resistant Candida albicans and C. auris Strains"

_jof, 2023, doi:10.3390/jof9080817_

Round 1
Reviewer 1 Report
This is an interesting study by Vargas-Casanova et al. describing for the first time that a peptide "RWQWRWQWR" in combination with ethanolic extracts from some parts of the plant "Bidens pilosa” have an important antifungal activity and low cytotoxicity. Besides, the authors found a synergy effect between the peptide and the fluconazole against a multiresistant yeast Candida auris.
The work is well written, well described, and the methodology used is appropriate. It also has a promising impact on the antifungal approaches.
However, there are a few issues that need to be addressed before acceptance.
1) The names of the microorganisms and the plant must be written fully the first time they are mentioned both in the abstract and in the body of the text.
2) The paragraph between lines 50 and 59 would look better in the discussion. Here, the authors could just mention the previous study and make clear the gap that still needs to be studied, which would make the introduction more interesting.
3) Line 60, please exclude the expression “On the other hand”,
4) the word “treatment” used several times is inappropriate and should be replaced, as there was no treatment but the exposure of the yeast to the compounds.
5) Line 254 ………. “(Table 1) [16]“ is confusing and needs to be clarified, leave in Table 1 only the results of the current study.
Author Response
This is an interesting study by Vargas-Casanova et al. describing for the first time that a peptide "RWQWRWQWR" in combination with ethanolic extracts from some parts of the plant "Bidens pilosa” have an important antifungal activity and low cytotoxicity. Besides, the authors found a synergy effect between the peptide and the fluconazole against a multiresistant yeast Candida auris.
The work is well written, well described, and the methodology used is appropriate. It also has a promising impact on the antifungal approaches.
However, there are a few issues that need to be addressed before acceptance.
Thank you for the comments and suggestions, the changes made are described below.
- The names of the microorganisms and the plant must be written fully the first time they are mentioned both in the abstract and in the body of the text.
Answer: According with reviewer recommendation the changes were made.
- The paragraph between lines 50 and 59 would look better in the discussion. Here, the authors could just mention the previous study and make clear the gap that still needs to be studied, which would make the introduction more interesting.
Answer: This information is an antecedent where we carried out a first screening of the antifungal activity of the peptide in different species of Candida, both sensitive and resistant to antifungals, where a broad spectrum of antifungal activity was evidenced and a first synergistic effect with fluconazole. This article is a continuation of the previously mentioned study.
Attend the reviewer comment part of this information was included in the discussion as follows:
Lines 339-343 Our results agree with previous studies [21], peptide R-1-R had antifungal activity against ATCC strains and clinical isolates of C. albicans, C. glabrata, C. krusei, C. auris, and C. tropicalis. Additionally, it was fungistatic and fungicidal and combined with FLC exhibited a synergistic antifungal effect against C. tropicalis 883 and C. krusei 6258 (resistant to FLC).
- Lanes 60, please exclude the expression “On the other hand”,
Answer: According with reviewer comment the change was made.
- the word “treatment” used several times is inappropriate and should be replaced, as there was no treatment but the exposure of the yeast to the compounds.
Answer: According with reviewer suggestion the changes were made as follows:
Line 161: …Serial dilutions were made, for peptides from…
Line 170: … concentration of peptide or extract that inhibited 50% of growth…
Line 174: … concentration of peptide or extract that…
Lines 598-599: … combinations of peptide and extract, the number…
Line 626: In the combinations of peptide and extract there was…
- Lane 254:“(Table 1) [16]“ is confusing and needs to be clarified, leave in Table 1 only the results of the current study
Answer: According with reviewer comment the change was made.

Reviewer 2 Report
In the work “Combining the peptide RWQWRWQWR and an ethanolic extract of Bidens pilosa enhances the activity against sensitive and resistant C. albicans and C. auris strains”, the authors studied the antifungal activity of peptide R-1-R, and its derivatives, ethanolic extract of B. pilosa, and their combination with fluconazole against C. albicans and C. auris, FLC-sensitive and -resistant strains.
My major concern is related to the experimental design and results’ presentation. However, I think the present manuscript is not suitable for publication, but it could be reconsidered after following major revisions. Minor questions are also reported.
Major comments:
I believe that all the considerations in the manuscript are based on an incorrect statement:
Line 165-167: The MIC was defined as the minimum concentration of treatment that inhibited 50% of growth compared to the control.
According to EUCAST and CLSI definitions, minimum inhibitory concentrations (MICs) are defined as the lowest concentration of an antimicrobial agent expressed in mg/L (μg/mL) which, under strictly controlled in vitro conditions, completely prevents visible growth of the test strain of an organism, and not “inhibited 50% of growth compared to the control”.
For publication on Journal of Fungi, I propose to include the following considerations.
- Report all results with the MIC values based on the standard definitions, or otherwise define the present results not as MIC but in another way, e.g., EC50 or other.
- I believe that the "MIC" values reported are too high for possible therapeutical applications (extract 500-1000 μg/ml, peptide 100-200 μg/ml).
- Figure 6: I disagree that the combinations show no cytotoxic effect against primary cell culture of fibroblast, the results report an average viability of 80% with a high standard deviation.
Minor comments:
- Write the name in full for all abbreviations the first time they appear, title, abstract, text, figure, etc.
- Line 24: replace “fluconazole” with “FLC” and review all abbreviations in the manuscript making them uniform, e.g., sabouraud agar (SA) or Sabouraud dextrose agar (SDA).
- Line 42: insert recent citations regarding the AMP/metabolite defence against all microorganism and not only C. albicans.
- Lines 44-46, 62: insert most recent citations.
- Line 47: delete “spp.”.
- Line 49: reference no. 15 is not appropriate, report the correct reference: Wakabayashi, H., Abe, S., Teraguchi, S., Hayasawa,H., and Yamaguchi, H., Inhibition of hyphal growth of azole-resistant strains of Candida albicans by triazole antifungal agents in the presence of lactoferrin-related compounds. Antimicrob. Agents Chemother.,42, 1587–1591 (1998).
- Lines 70-75: check each reference in the text and/or in the reference list and their correspondence; respect the numerical order.
- Line 81: delete “and” before cervix.
- Lines 87-89: revise the sentence, as written is not clear. Is 1,562 the number 1.562? Refers “(44,46)” to references 44 and 46? If so, there is no match between reference and text.
- Line 100: brackets are missing.
- Line 101: is “100-well microplates” correct?
- Lines 120, 121, 126: replace “hours” with “h”.
- Lines 138, 161, 174, 183, 195, 206, 233: replace “x” with multiplication sign “×”.
- Line 139: delete 0.1% before formic acid.
- From line 140: standardize the spaces before “%” and before and after “-” and “< >”.
- Line 147: replace “-3kV” with “-3 kV”.
- Lines 164, 355, 407: replace “y” with “and”.
- Lines 170, 636: delete the period after control.
- Lines 181-183, 191-192: revise the sentences, as written is not clear.
- Line 183: replace “103” with “103”.
- Line 198: move reference 53 to line 194 after “instructions”.
- Lines 207, 258: revise correspondence between references and text.
- Line 247: “versus” in italics.
- Lines 349, 357: delete the period at the end of the title.
- Line 276: [5K]-K-1-K is the peptide where all the Arg residues have been replaced with Lys, not just the fifth one, review the previous sentence or give another example.
- Lines 289-294, 328-335: revise the sentences, as written is not clear.
- Line 299: replace “gram” with “Gram”.
- Line 319: delete “and this peptide”.
- Line 326: add “positive” before charge.
- Line 328: why was the lower concentration of extract and peptide doubled? 31.25 μg/ml and 3.12 μg/ml for 15.62 μg/ml and 1.56 μg/ml, respectively.
- Line 334: what are PAMs?
- Table 1: specify the meaning of the highlighted lines.
- Line 368: revise text format of “the reference standard”.
- Table 2: insert table on one page. Add “of” after extracts in the title.
- Line 387: move “(CLSI method)” after MIC values.
- Line 389: “in silico” in italics.
- Lines 415-416: what are the fungicidal concentrations?
- Line 418: two, not four strains.
- Line 431: replace “for” with “of”.
- Line 498: the palindromic peptide does not enhance the activity of FLC, its effect is additive, not synergic.
- Line 501: add reference regarding synergistic or additive effect of peptide and FLC or AmB.
- Line 545: “all” is not correct.
- Line 558: replace “S1-S4” with “S12-S15”. In these figures, detail the caption of the figures and replace “LfcinB (21-15)Pal” with “R-1-R”.
- Figure 6: report after its description in the text.
- Line 656: delete the period after viability.
- Figures S1-S11 are not cited in the text.
- Standardize the list of references (e.g., author, microorganism name in italics, doi, …).
Author Response
In the work “Combining the peptide RWQWRWQWR and an ethanolic extract of Bidens pilosa enhances the activity against sensitive and resistant C. albicans and C. auris strains”, the authors studied the antifungal activity of peptide R-1-R, and its derivatives, ethanolic extract of B. pilosa, and their combination with fluconazole against C. albicans and C. auris, FLC-sensitive and -resistant strains.
My major concern is related to the experimental design and results’ presentation. However, I think the present manuscript is not suitable for publication, but it could be reconsidered after following major revisions. Minor questions are also reported.
ANSWER
Thank you for the considerations and suggestions, in order to clarify the reviewer observations, we make the following comments and changes.
Major comments:
I believe that all the considerations in the manuscript are based on an incorrect statement:
Line 165-167: The MIC was defined as the minimum concentration of treatment that inhibited 50% of growth compared to the control.
According to EUCAST and CLSI definitions, minimum inhibitory concentrations (MICs) are defined as the lowest concentration of an antimicrobial agent expressed in mg/L (μg/mL) which, under strictly controlled in vitro conditions, completely prevents visible growth of the test strain of an organism, and not “inhibited 50% of growth compared to the control”.
For publication on Journal of Fungi, I propose to include the following considerations.
- Report all results with the MIC values based on the standard definitions, or otherwise define the present results not as MIC but in another way, e.g., EC50 or other.
Answer: The interpretation of results to determination of the MIC, was carried out according to the definition reported by the CLSI M27A4 document: “the lowest concentration of an antimicrobial agent that causes and specified reduction in visible growth of a microorganism in agar or broth dilution susceptibility test. NOTE: The magnitude of reduction in visible growth is assessed using the following numerical scale: 0, optically clear; 1, slightly hazy; 2, prominent decrease (≈50%) in visible growth; 3, slight reduction in visible growth; and 4, no reduction in visible growth”.
“The document also establishes that to azoles, end points are typically less defined than those described for amphotericin B, which may contribute to a significant source of variability. Applying a less stringent end point, an approximately 50% reduction in growth relative to the antifungal agent-free growth control, has improved interlaboratory agreement and distinguishes between presumed susceptible and resistant isolates”.
Since our study focused on the combination of the peptide and/or extract with fluconazole and in order made our results to be comparable, we based ourselves on the interpretation of results for azoles, where, as seen above, the MIC is determined with the score 2 (prominent "≈50%" decrease in visible growth).
Reference: CLSI. Reference Method for Broth Dilution Antifungal Susceptibility Testing of Yeast. 4th ed. CLSI standard M27. Wayne, PA: Clinical and Laboratory Standards Intitute; 2017.
According to the previosly discussed paragraph, the follows sentence was included in line 406
….(as described by CLSI method [54-56])
- I believe that the "MIC" values reported are too high for possible therapeutical applications (extract 500-1000 μg/ml, peptide 100-200 μg/ml).
Answer: In order to clarify this topic, the follows paragraph was included in Lanes 344-354:
The antifungal activity of peptide R-1-R against C. albicans was comparable to that reported for LFB (MIC 200->6400 μg/mL) and LfcinB (MIC 0.8-400 μg/mL). Added, the antifungal activity of the peptide R-1-R is in agreement with that previously reported for short peptides derived from LfcinB against other C. albicans strains: Peptide 2 (FKCRRWQWRM; MIC 17.3-17.5 μM), bLF (or LFB 17–30) (FKCRRWQWRMKKLGA; MIC 5–10 μM), Lfcin B-20 (LfcinB 18–37) (FKCRRWQWRMKKLGA; MIC 8 μg/mL), Lfcin B-18-42 (MIC 100 μg/mL); Lfcin B-9 (RRWQWRMKK; MIC 25000–32000 μg/mL), Peptide 5 (RWQWRM; MIC 500 μM), and Peptide 3 (GAPSITCVRRAF; MIC 635 μM). The peptide R-1-R also exhibited similar antifungal activity to other palindromic sequences rich in Arg and/ or Trp residues in some C. albicans strains, as follows: IRIRIRIR, KWKWWKWK, KWKWKWKW [21].
Answer: To the extract, the antifungal activity reported in the present study was similar and even higher than that obtained by other authors, this is discussed in the lines 387 to 392. Although there is no regulation regarding cut-off points for the antifungal activity of peptides or plant extracts, based on reviews of the literature, according to Alves, et al, both peptide and extract exhibited weak to moderate activity (Lines 443-444)
Reference: Alves, D.D.N.; Ferreira, A.R.; Duarte, A.B.S.; Melo, A.K.V.; De Sousa, D.P.; Castro, R.D. De Breakpoints for the Classification of Anti- Candida Compounds in Antifungal Screening. Biomed Res. Int. 2021, 2021, doi:10.1155/2021/6653311
-Figure 6: I disagree that the combinations show no cytotoxic effect against primary cell culture of fibroblast, the results report an average viability of 80% with a high standard deviation.
Answer: Thank you very much for the observation, according to the comment, the results were extended from 3 technical replicates to 3 biological replicates and the high standard deviations have been corrected, confirming the viability results around 80%. Although cut-off points on cell viability have not been established, authors report no toxicity from viability percentages of between 70% and 80%, and indicate that these percentages could be safe for the host.
References
Barut, B.; Sari, S.; Sabuncuoğlu, S.; Özel, A. Azole Antifungal Compounds Could Have Dual Cholinesterase Inhibitory Potential According to Virtual Screening, Enzyme Kinetics, and Toxicity Studies of an Inhouse Library. J. Mol. Struct. 2021, 1235, 130268, doi:10.1016/J.MOLSTRUC.2021.130268.
Cannella, V.; Altomare, R.; Chiaramonte, G.; Di Bella, S.; Mira, F.; Russotto, L.; Pisano, P.; Guercio, A. Cytotoxicity Evaluation of Endodontic Pins on L929 Cell Line. Biomed Res. Int. 2019, 2019, doi:10.1155/2019/3469525.
López-García, J.; Lehocký, M.; Humpolíček, P.; Sáha, P. HaCaT Keratinocytes Response on Antimicrobial Atelocollagen Substrates: Extent of Cytotoxicity, Cell Viability and Proliferation. J. Funct. Biomater. 2014, 5, 43, doi:10.3390/JFB5020043.
ISO 10993-5:2009 Biological Evaluation of Medical Devices. Part 5: Tests for In Vitro Cytotoxicity. International Organization for Standardization; Geneva, Switzerland: 2009
Minor comments:
- Write the name in full for all abbreviations the first time they appear, title, abstract, text, figure, etc.
Answer: The changes were made
- Line 24: replace “fluconazole” with “FLC” and review all abbreviations in the manuscript making them uniform, e.g., sabouraud agar (SA) or Sabouraud dextrose agar (SDA).
Answer: The changes were made
- Line 42: insert recent citations regarding the AMP/metabolite defence against all microorganism and not only C. albicans.
Answer: The references were added
- Lines 44-46, 62: insert most recent citations.
Answer: The references were added to AMPs, but regarding recent information on lactoferricin the information is oriented towards synthetic peptides derived from it.
- Line 47: delete “spp.”.
Answer: The change was made
- Line 49: reference no. 15 is not appropriate, report the correct reference: Wakabayashi, H., Abe, S., Teraguchi, S., Hayasawa,H., and Yamaguchi, H., Inhibition of hyphal growth of azole-resistant strains of Candida albicans by triazole antifungal agents in the presence of lactoferrin-related compounds. Antimicrob. Agents Chemother.,42, 1587–1591 (1998).
Answer: The change was made.
- Lines 70-75: check each reference in the text and/or in the reference list and their correspondence; respect the numerical order.
Answer: The references were checked y adjusted.
- Line 81: delete “and” before cervix.
Answer: The change was made.
- Lines 87-89: revise the sentence, as written is not clear. Is 1,562 the number 1.562? Refers “(44,46)” to references 44 and 46? If so, there is no match between reference and text.
Answer: The number corresponds to 1.562, the change was made.
References were adjusted.
- Line 100: brackets are missing.
Answer: The brackets were added.
- Line 101: is “100-well microplates” correct?
Answer: This statement is correct, these plates are specific for the equipment Bioscreen C, which contain 100 wells.
- Lines 120, 121, 126: replace “hours” with “h”.
Answer: The changes were made.
- Lines 138, 161, 174, 183, 195, 206, 233: replace “x” with multiplication sign “×”.
Answer: The changes were made.
- Line 139: delete 0.1% before formic acid.
Answer: The change was made.
- From line 140: standardize the spaces before “%” and before and after “-” and “< >”.
Answer: The changes were made.
- Line 147: replace “-3kV” with “-3 kV”.
Answer: The change was made.
- Lines 164, 355, 407: replace “y” with “and”.
Answer: The changes were made.
- Lines 170, 636: delete the period after control.
Answer: The changes were made.
- Lines 181-183, 191-192: revise the sentences, as written is not clear.
Answer: The changes were made.
- Line 183: replace “103” with “103”.
Answer: - Line 198: move reference 53 to line 194 after “instructions”.
Answer: The changes were made.
- Lines 207, 258: revise correspondence between references and text.
Answer: After reviewing the references, on line 207, the reference was removed and on line 258 the reference does correspond to the text.
- Line 247: “versus” in italics.
The change was made.
- Lines 349, 357: delete the period at the end of the title.
Answer: The changes were made in the lines 249 and 357
- Line 276: [5K]-K-1-K is the peptide where all the Arg residues have been replaced with Lys, not just the fifth one, review the previous sentence or give another example.
Answer: According with reviewer comment, the code was modified by K-1-K
- Lines 289-294, 328-335: revise the sentences, as written is not clear.
Answer: Sentences were checked and adjusted.
- Line 299: replace “gram” with “Gram”.
Answer: The change was made.
- Line 319: delete “and this peptide”.
Answer: The change was made
- Line 326: add “positive” before charge.
Answer: The change was made.
- Line 328: why was the lower concentration of extract and peptide doubled? 31.25 μg/ml and 3.12 μg/ml for 15.62 μg/ml and 1.56 μg/ml, respectively.
Answer: Lanes 242-243: The concentrations evaluated were B. pilosa extract (31.25-2000 µg/mL) and peptide (3.12-200 μg/mL). These concentrations are the same used in the antifungal activity assays.
- Line 334: what are PAMs?
Answer: The abbreviation corresponds to Antimicrobial Peptides (AMP), the change was made throughout the document
- Table 1: specify the meaning of the highlighted lines.
Answer: Highlighted lines correspond to concentrations similar or less than the concentrations of the original R-1-R peptide. But it is decided not to make these lines since that information is described in the text.
- Line 368: revise text format of “the reference standard”.
Answer: The format of the text was adjusted
- Table 2: insert table on one page. Add “of” after extracts in the title.
Answer: The table will be adjusted by de editors.
Answer: The word “of” was added
- Line 387: move “(CLSI method)” after MIC values.
Answer: The change was made
- Line 389: “in silico” in italics.
Answer: The change was made
- Lines 415-416: what are the fungicidal concentrations?
Answer: This information was added to the text: (C. albicans SC5314: 50 μg/mL (34 μM), C. albicans 256: 100 μg/mL (67 μM), C. auris 400 μg/mL (270 μM) and C. auris >400 μg/mL (>270 μM)).
- Line 418: two, not four strains.
Answer: Corresponds to the four study strains, a point is added to give clarity to the paragraph.
- Line 431: replace “for” with “of”.
Answer: The change was made
- Line 498: the palindromic peptide does not enhance the activity of FLC, its effect is additive, not synergic.
Answer: Although the effect is additive, it is evident that with the peptide it is possible to reduce the MIC of the FLC, which is especially relevant in the resistant strain C. albicans 256, which from a resistant phenotype becomes susceptible to FLC. The key activity of LficnB-derived peptides, when combined with FLC, has been previously demonstrated by Lupetti et al, where the synergistic effect was obtained either by adding FLC first and then the peptide hLF(1-11), or by adding peptide and FLC at the same time, but no synergy is observed if FLC was added first. This indicates that the peptide is an initiator and the FLC an effector of the antifungal activity.
Reference: Lupetti, A.; Paulusma-Annema, A.; Welling, M.M.; Dogterom-Ballering, H.; Brouwer, C.P.J.M.; Senesi, S.; Van Dissel, J.T.; Nibbering, P.H. Synergistic Activity of the N-Terminal Peptide of Human Lactoferrin and Fluconazole against Candida Species. Antimicrob. Agents Chemother. 2003, 47, 262–267, doi:10.1128/aac.47.1.262-267.2003.
- Line 501: add reference regarding synergistic or additive effect of peptide and FLC or AmB.
Answer: The reference was added, and AmB is changed to Caspofungin (CAS)
- Line 545: “all” is not correct.
Answer: The phrase: “except for [3A]-R-1-R, which was additive” was added, to specify the result
- Line 558: replace “S1-S4” with “S12-S15”. In these figures, detail the caption of the figures and replace “LfcinB (21-15)Pal” with “R-1-R”.
Answer: The changes were made in the text and the figures.
- Figure 6: report after its description in the text.
Answer: Figure 6 is mentioned in the following paragraph, line 666 - Line 656: delete the period after viability.
- Figures S1-S11 are not cited in the text.
Answer: Figure S1-S11 are cited in the numeral: “3.2.1. Phytochemical Analysis” Lines 359-361
- Standardize the list of references (e.g., author, microorganism name in italics, doi, …).
Answer: The references were adjusted

Round 2
Reviewer 2 Report
I thank the authors for the explanations, but I think there are still some aspects to clarify.
“The document also establishes that to azoles, end points are typically less defined than those described for amphotericin B, which may contribute to a significant source of variability. Applying a less stringent end point, an approximately 50% reduction in growth relative to the antifungal agent-free growth control, has improved interlaboratory agreement and distinguishes between presumed susceptible and resistant isolates”.
I agree with this statement, but I don't find it correct to extend this to other compounds as peptides and/or extract, for which the MIC value is calculated as the lowest concentration which completely prevents visible growth.
The authors motivated this choice in order to compare their results “focused on the combination of the peptide and/or extract with fluconazole”, but this would make comparisons with other studies difficult.
In this regard, the authors correctly defined how the MIC was calculated in this paper (lines 169-171), but do the MICs reported in the examples from lines 344 to 350 (base of the comparison with R-1-R) refer to the standard definition or to that of azoles?
Furthermore, the same authors cited Angelini et al. [28] (lines 405-407), emphasizing that the MIC values were calculated “as described by CLSI method [54–56]”; indeed, Angelini et al. [28] reported “MIC end-points were defined as the lowest concentration of either B. pilosa extracts or ciprofloxacin that totally inhibited bacterial growth”.
In conclusion, whether the authors deemed it appropriate to calculate the MIC value for peptides, extracts and/or their combinations based on the interpretation of results for azoles “in order made our results to be comparable”, this could have a scientific soundness limited to this study, but a comparison with the results of other studies, also cited by the same authors, in which the MIC assumes a different meaning is not possible.
Minor comments are highlighted in the attached pdf.

Author Response
Dear reviewer,
Comments and Suggestions for Authors
I thank the authors for the explanations, but I think there are still some aspects to clarify.
“The document also establishes that to azoles, end points are typically less defined than those described for amphotericin B, which may contribute to a significant source of variability. Applying a less stringent end point, an approximately 50% reduction in growth relative to the antifungal agent-free growth control, has improved interlaboratory agreement and distinguishes between presumed susceptible and resistant isolates”.
I agree with this statement, but I don't find it correct to extend this to other compounds as peptides and/or extract, for which the MIC value is calculated as the lowest concentration which completely prevents visible growth.
Realmente nos hemos esforzado en generar nuestro protocolo estándar, debido a que no hay uno, si conoce algún protocolo que estemos desconociendo, le agradeceríamos lo comparta, lo mas cercano que hemos encontrado se encuentra en el siguiente enlace que es para bacterias.
We have really worked hard to generate our standard protocol, since there isn't one, if you know of any protocol that we are unaware of, we would appreciate it if you share it, the closest we have found is in the following link which is for bacteria.
http://cmdr.ubc.ca/bobh/method/modified-mic-method-for-cationic-antimicrobial-peptides/
Es importante tener en cuenta, que no podemos extrapolar a hongos todos los protocolos de bacterias, teniendo en cuenta las grandes diferencias en su biología. En bacterias es claro que efectos bacteriostáticos no son deseables para tratar enfermedades infecciosas, sin embargo, en hongos sí. Esto motivado por la alta toxicidad de los compuestos que presentan una actividad fungicida. Por esto los abordajes que potencien el uso del fluconazol juntando actividades fungistáticas consideramos que es un camino que eventualmente podría ser exitoso.
It is important to bear in mind that we cannot extrapolate all bacterial protocols to fungi, taking into account the great differences in their biology. In bacteria it is clear that bacteriostatic effects are not desirable to treat infectious diseases, however, in fungi they are. This is motivated by the high toxicity of the compounds that have a fungicidal activity. For this reason, we believe that approaches that promote the use of fluconazole by combining fungistatic activities are a path that could eventually be successful.
The authors motivated this choice in order to compare their results “focused on the combination of the peptide and/or extract with fluconazole”, but this would make comparisons with other studies difficult.
Estamos de acuerdo, en el que este camino está en construcción, y que es una manera alternativa de buscar actividad antifúngica. por este motivo incluimos fotos del claro efecto en membrana que produce el péptido y la combinación con el extracto que corrobora, que las MIC, si están afectando a las levaduras a nivel de membrana y pared y que a estas mismas concentraciones tienen bajos efectos hemolíticos y citotóxicos. Estos resultados son positivos y muy reproducibles y hacen que los tratamientos evaluados conserven su potencial uso como antifúngicos.
We agree that this pathway is under construction, and that it is an alternative way to search for antifungal activity. For this reason, we include photos of the clear effect on the membrane produced by the peptide and the combination with the extract that corroborates that the MICs are affecting the yeasts at the membrane and wall level and that at these same concentrations they have low hemolytic effects and cytotoxic. These results are positive and very reproducible and make the evaluated treatments retain their potential use as antifungals.
Además, con motivo a las dudas generadas en la interpretación de la actividad antifúngica, adicionamos para esta versión revisada del artículo, otros resultados que muestra efectos a nivel de la permeabilización de la membrana con ioduro de propidio, de los tratamientos que estamos sometiendo a publicación.
In addition, due to the doubts generated in the interpretation of the antifungal activity, we add for this revised version of the article, other results that show effects at the level of membrane permeabilization with propidium iodide, of the treatments that we are submitting for publication
In this regard, the authors correctly defined how the MIC was calculated in this paper (lines 169-171), but do the MICs reported in the examples from lines 344 to 350 (base of the comparison with R-1-R) refer to the standard definition or to that of azoles?
Estamos basándonos en la definición estándar del CLSI que es para azoles (grupo con el mayor número de antifúngicos disponibles). Y molécula a la cual es resistente la cepa PUJ256, en la cual vemos un fenotipo sensible en el momento de la combinación con el péptido. Teniendo en cuenta que para anfotericina B y equinocandinas que son fungicidas es diferente. Esto citado en el mismo documento del CLSI. Esta es la mejor forma que hemos conseguido teniendo en cuenta que estamos combinando diferentes abordajes, valuación de fluconazol, péptidos y extractos de plantas.
We are basing ourselves on the CLSI standard definition, which is for azoles (the group with the largest number of antifungals available). And a molecule to which the PUJ256 strain is resistant, in which we see a sensitive phenotype at the time of combination with the peptide. Bearing in mind that for amphotericin B and echinocandins, which are fungicides, it is different. This is cited in the same CLSI document. This is the best way we have achieved considering that we are combining different approaches, fluconazole titration, peptides and plant extracts.
Incluso para la evaluación de extractos de plantas, no hay una normativa, se sigue publicando con métodos que evalúan la difusión en agar, metodología con la cual no nos sentimos cómodos, por las limitaciones que presenta.
Even for the evaluation of plant extracts, there is no regulation, methods that evaluate diffusion in agar continue to be published, a methodology with which we do not feel comfortable, due to the limitations it presents.
Furthermore, the same authors cited Angelini et al. [28] (lines 405-407), emphasizing that the MIC values were calculated “as described by CLSI method [54–56]”; indeed, Angelini et al. [28] reported “MIC end-points were defined as the lowest concentration of either B. pilosa extracts or ciprofloxacin that totally inhibited bacterial growth”.
Estamos dando algún ejemplo, de trabajos publicados con el extracto que estamos trabajando, sin embargo, como explicamos anteriormente, los protocolos de bacterias no son extrapolables al 100% en hongos. Ellos no usan un antifúngico para el caso de hongos, solo ciprofloxacina, la cual actúa como bactericida.
We are giving some examples of published works with the extract we are working on, however, as we explained previously, the bacterial protocols cannot be 100% extrapolated to fungi. They do not use an antifungal in the case of fungi, only ciprofloxacin, which acts as a bactericide.
In conclusion, whether the authors deemed it appropriate to calculate the MIC value for peptides, extracts and/or their combinations based on the interpretation of results for azoles “in order made our results to be comparable”, this could have a scientific soundness limited to this study, but a comparison with the results of other studies, also cited by the same authors, in which the MIC assumes a different meaning deis not possible.
La solidez científica se muestra en resultados como en el retorno a un perfil de susceptibilidad de la cepa resistente al fluconazol, y los observados por microscopia donde muestran que la permeabilidad de membrana es alterada por los tratamientos utilizados.
La investigación no puede estar limitada por los protocolos estandarizados, hay que ajustarse y proveer resultados que muestren otras formas de alternativas de evaluación, que esperamos con el tiempo sean utilizadas por otros grupos.
The scientific solidity is shown in results such as the return to a susceptibility profile of the strain resistant to fluconazole, and those observed by microscopy where they show that membrane permeability is altered by the treatments used.
Research cannot be limited by standardized protocols, it is necessary to adjust and provide results that show other forms of evaluation alternatives, which we hope will be used by other groups over time.
Minor comments are highlighted in the attached pdf
Was made.
Answer:
The value of CMI, is not added in the figure 2c, because the effect of the MIC was similar equally fungistatic between 0.5 MIC and 2 MIC, and for this particular case, only 2 MIC was left to show that there is no fungicidal effect at the highest concentration evaluated.
Line 518, The statement is made according to the results obtained especially in the resistant strain C. albicans 256, where, as we have previously reported, the peptide is necessary as an initiator for the FLC to be able to exert an effector activity to kill the yeast.

Round 3
Reviewer 2 Report
I thank the authors for their explanations.
Minor comments are highlighted in the attached pdf.

Author Response
Dear reviewer
Minor revisions were made
Lines 357 -360
The units of the MICs are those reported by the authors in the original articles, we prefer not to modify the data they show, for this reason and to make comparisons with the bibliography, we show our results (table 1) both in µg/mL as in µM.
